# Metastable Dynamics of Chain-of-Thought Reasoning: Provable Benefits of Search, RL and Distillation

**Juno Kim** [1]  **Denny Wu** [2]  **Jason D. Lee** [3]  **Taiji Suzuki** [1]

## Abstract

A key paradigm to improve the reasoning capabilities of large language models (LLMs) is to allocate more inference-time compute to search against a verifier or reward model. This process can then be utilized to refine the pretrained model or distill its reasoning patterns into more efficient models. In this paper, we study inference-time compute by viewing chain-of-thought (CoT) generation as a metastable Markov process: easy reasoning steps (e.g., algebraic manipulations) form densely connected clusters, while hard reasoning steps (e.g., applying a relevant theorem) create sparse, low-probability edges between clusters, leading to phase transitions at longer timescales. Under this framework, we prove that implementing a search protocol that rewards sparse edges improves CoT by decreasing the expected number of steps to reach different clusters. In contrast, we establish a limit on reasoning capability when the model is restricted to local information of the pretrained graph. We also show that the information gained by search can be utilized to obtain a better reasoning model: (1) the pretrained model can be directly finetuned to favor sparse edges via policy gradient methods, and moreover (2) a compressed *metastable representation* of the reasoning dynamics can be distilled into a smaller, more efficient model.

## 1. Introduction

Pretraining and inference constitute two distinct computational phases in large language models (LLMs). The pretraining phase, during which the model learns from vast amounts of text data through next-token prediction (Rad-

[1]University of Tokyo and RIKEN AIP [2]New York University and Flatiron Institute [3]Princeton University. Correspondence to: Juno Kim <junokim@berkeley.edu>.

*Proceedings of the 42$^{nd}$ International Conference on Machine Learning*, Vancouver, Canada. PMLR 267, 2025. Copyright 2025 by the author(s).

ford et al., 2018), is well known for its high computational demands, and its scaling behavior has been extensively studied (Kaplan et al., 2020; Hoffmann et al., 2022; Dubey et al., 2024). On the other hand, inference (running the trained model to generate responses) was traditionally considered computationally inexpensive, until a recent paradigm shift demonstrating that model reasoning capabilities can drastically improve by allocating more computational resources during inference time (Jaech et al., 2024; Guo et al., 2025; Kimi et al., 2025). Hence it is crucial to understand the advantages scaling inference computation can provide beyond those achieved through pretraining (Jones, 2021; Snell et al., 2024; Wu et al., 2024).

Reasoning LLMs follow the chain-of-thought (CoT) (Nye et al., 2021; Wei et al., 2022) format where intermediate reasoning steps are iteratively generated before arriving at a final answer. Various reinforcement learning (RL) based approaches (Bai et al., 2022) have been proposed to improve CoT quality at inference time, such as process reward modeling (Lightman et al., 2023; Uesato et al., 2022), Monte-Carlo Tree Search (MCTS) (Silver et al., 2018; Feng et al., 2023b; Trinh et al., 2024; Xie et al., 2024), and data self-generation (Zelikman et al., 2022; Kumar et al., 2024). Theoretically, the benefit of (sufficiently long) CoT has been studied in terms of expressive power and statistical efficiency (Merrill & Sabharwal, 2023; Li et al., 2024b; Kim & Suzuki, 2024; Wen et al., 2024).

Motivated by the discrete and sequential nature of CoT, we follow Xu et al. (2019); Sanford et al. (2024a); Abbe et al. (2024); Besta et al. (2024) and consider learning on graphs as an ideal abstraction of complex reasoning tasks. We model pretraining as the process of discovering the graph structure, or the *linguistic* (world) model, upon which a *reasoning* (inference) component is implemented to search for a valid path between states (at a high level, this division parallels the System 1 vs. System 2 distinction discussed in Kahneman (2011); Xiang et al. (2025)). Building on the observation that intermediate reasoning steps vary in difficulty, we assume the underlying graph consists of dense clusters connected by sparse, low-probability edges representing "hard" reasoning steps. We further model CoT generation as a Markov process and characterize hitting/escape times

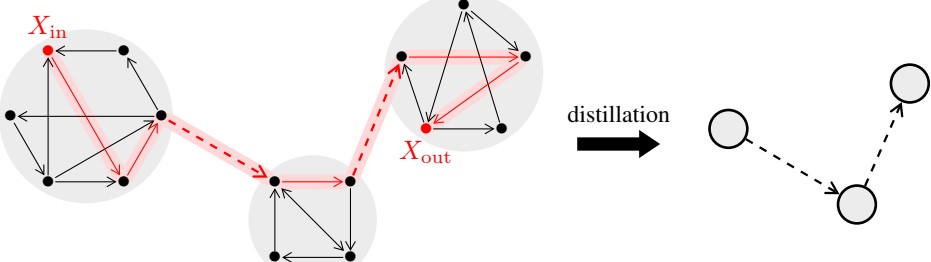

*Figure 1. (Left)* Example of metastable graph with three clusters. Each state represents a logical assertion and edges correspond to reasoning steps. Solid and dashed arrows indicate easy (within-cluster) and hard (inter-cluster) reasoning steps, respectively. The goal of the reasoner is to retrieve a valid CoT path from $X_{\text{in}}$ to $X_{\text{out}}$ (highlighted). Search aims to use CoT generated from the pretrained model to explore the linguistic model and identify hard steps, which can then be used to fine-tune the pretrained model via RL to improve its generation. *(Right)* The coarse-grained dynamics of CoT at long timescales can be represented by a meta-chain on the set of clusters and distilled into a smaller model, which can generate reasoning paths more efficiently.

by leveraging *metastability theory* (Bovier et al., 2002; Betz & Le Roux, 2016), which describes systems with multiple locally stable states separated by high energy barriers, leading to a timescale separation between local and global transitions (e.g., a reasoner may become stuck at a critical reasoning step for an extended period). Our toy model captures key phenomena observed in the training of reasoning LLMs:

- *Benefit of search and RL.* Inference-time search elicits reasoning capabilities beyond pretraining (Jones, 2021; Yao et al., 2024; Snell et al., 2024). Roughly speaking, running search on the pretrained graph identifies important reasoning steps, and then RL can improve the base linguistic model by modifying the graph and reweighting the corresponding transition probabilities.

- *Benefit of distillation.* Reasoning patterns can be distilled into a smaller model (Hsieh et al., 2023; Gandhi et al., 2024; Guo et al., 2025). By training on curated CoT data of the larger model, we can efficiently represent the reasoning dynamics with a much smaller meta-chain that compresses the dense clusters (representing "easy" steps).

## 1.1. Our Contributions

We study the metastable Markov process underlying CoT generation (see Figure 2) which provides insights into the roles of pretraining, search, RL, and distillation. Our contributions are summarized as follows.

- In Section 2, we introduce a perturbed Markov chain model for CoT reasoning that differentiates between easy and hard reasoning steps through a dense-sparse structure. We develop a quantitative analysis of its metastable dynamics over long timescales by deriving tight bounds on the expected hitting times of target states.

- In Section 3, we demonstrate that inference-time search

based on intrinsic reward improves hitting times by identifying key reasoning steps, whose generation can be enhanced directly or by fine-tuning the base model with RL. Moreover, optimization guarantees for pretraining and RL (PPO-Clip) are provided for a simple softmax model.

- In Section 4, we show that a compressed version of the CoT dynamics can be distilled to a smaller model by only learning the macroscopic cluster transitions. We prove that this representation efficiently maps out paths through clusters while preserving essential dynamical quantities of the original chain.

- Finally, in Section 5 we prove that large test time compute (unbounded search) is necessary to solve a computational version of the path-finding task, by introducing a new statistical query (SQ) complexity measure that accounts for additional information the learner can access (e.g., CoT path, local search data).

All proofs are deferred to the appendix. A discussion of additional related works is provided in Appendix A. Metastable dynamics and hitting times are studied in Appendices B-C, optimization dynamics are analyzed in Appendix D, and learning-theoretic lower bounds are given in Appendix E.

## 2. Metastable Dynamics and Reasoning

### 2.1. CoT as Markov Chains

Our key insight to understanding inference-time search is to frame CoT reasoning as a metastable Markov process over an underlying linguistic model. Each state represents a logical assertion (e.g., a sentence or mathematical expression rather than a single token), and state transitions correspond to reasoning steps. The model distinguishes between **easy/trivial reasoning steps**, which form dense local clusters of roughly equivalent meaning, and **hard/crucial reasoning steps**, which form sparse connections between

clusters of small probability $O(\varepsilon)$. Reasoning paths sampled from this process typically spend a long time in each cluster before making a nontrivial jump to another cluster. This leads to a dynamical separation between fast and slow timescales, which we quantitatively study by tuning the degree $\varepsilon$ of perturbation.

The setup is formalized as follows. Let $X^\varepsilon = (X_t^\varepsilon)_{t \geq 0}$ be a perturbed family of discrete-time stationary Markov chains on a (large but finite) state space $S$ with transition kernel $p^\varepsilon$, such that $p^\varepsilon$ uniformly converges to $p^0$ as $\varepsilon \to 0$. We assume $X^\varepsilon$ is recurrent for all $\varepsilon \geq 0$ and irreducible for all $\varepsilon > 0$; also, $X^0$ is reducible and decomposes $S$ into $K$ disjoint $p^0$-ergodic components $C_1, \cdots, C_K$. We set $M := \max_k |C_k|$ and assume that $\min_k |C_k| = \Theta(M)$ and $K \leq \mathrm{poly}(M)$. Moreover, we denote the stochastic complement of $p^\varepsilon$ corresponding to $C_k$ by the matrix $\mathbf{S}_{kk}^\varepsilon$; see Appendix B for definitions. The stationary distributions of $p^\varepsilon, \mathbf{S}_{kk}^\varepsilon$ are denoted by $\pi^\varepsilon, \pi_k^\varepsilon$ and we set $\mu_k := \pi_k^0$.

**Assumption 1** (dense clusters). *For each $\mathbf{S}_{kk}^\varepsilon$, the pseudo-spectral gap $\gamma^\dagger(\mathbf{S}_{kk}^\varepsilon) \geq \gamma > 0$ and the stationary measure $\pi_k^\varepsilon$ satisfies $\pi_k^\varepsilon(x) = \Theta(1/M)$ for all $x \in C_k$.*

We give verifiable conditions on the unperturbed kernel $p^0$ which guarantee Assumption 1 in Proposition B.8.

We further denote $E_0 = \mathrm{supp}\, p^0$ and assume that $E = \mathrm{supp}\, p^\varepsilon$ is fixed for all $\varepsilon > 0$. A reasoning path $X_{0:T}$ is termed *valid* if $(X_{t-1}, X_t) \in E$ for all $t \in [T]$. The set of sparse edges is denoted by $E_s = E \setminus E_0$.

**Assumption 2** (sparse edges). *There are at most $d_{\mathrm{out}}$ sparse edges from each of at most $n_{\mathrm{out}}$ sources in $C_k$, and there is at most one sparse edge between any two distinct clusters, with at least one sparse edge from each cluster. Moreover, $p^\varepsilon(y|x) \propto \varepsilon$ for each $(x, y) \in E_s$ with proportionality constant bounded above and below w.r.t. $M, K$, and $p^\varepsilon(z|x)$ for $(x, z) \in E_0$ all decrease proportionally with $\varepsilon$.*

## 2.2. Reasoning Task

The reasoner is given a pair of input and output states $(X_{\mathrm{in}}, X_{\mathrm{out}})$ sampled from a distribution $\mathcal{D}$ on $S \times S$. The goal of the reasoner is to find a valid path from $X_{\mathrm{in}}$ to $X_{\mathrm{out}}$. We are thus interested in the hitting time of CoT generation to understand inference-time computation. The overall difficulty of the task is measured by the minimum number of hard reasoning steps needed to reach $X_{\mathrm{out}}$ from $X_{\mathrm{in}}$; longer reasoning chains will require more sparse transitions. We assume the average difficulty of the task is lower bounded:

**Assumption 3.** *For $(X_{\mathrm{in}}, X_{\mathrm{out}}) \sim \mathcal{D}$ and any valid path $X_{0:T}$ with $X_0 = X_{\mathrm{in}}$ and $X_T = X_{\mathrm{out}}$, it holds that*

$$\mathbb{E}_{\mathcal{D}}[\min |X_{0:T} \cap E_s|] = \Omega(K).$$

Note that $X_{\mathrm{out}}$ is already known in our setting. For example, for theorem proving, $X_{\mathrm{in}}$ is the problem statement and $X_{\mathrm{out}}$

is the QED symbol; or when asked a "why" question, $X_{\mathrm{out}}$ could be the conclusion, "That is why..." Nonetheless, for many reasoning problems the answer is unknown and must be deduced or computed. We incorporate this aspect by introducing a 'logical computation' task in Section 5.

## 2.3. Metastable Dynamics

The *hitting time* and *return time* of $X^\varepsilon$ to a set $A \subseteq S$ are defined as $\tau_A^\varepsilon = \inf\{t \geq 0 : X_t^\varepsilon \in A\}$, $\bar{\tau}_A^\varepsilon = \inf\{t > 0 : X_t^\varepsilon \in A\}$, respectively. Probabilities and expectations conditioned on the initial state $x$ are denoted as $\mathbb{P}_x, \mathbb{E}_x$, etc.

In the context of perturbed Markov chains, a subset $\mathcal{M} \subset S$ is defined as a *metastable system* (Bovier et al., 2002) if

$$\lim_{\varepsilon \to 0} \sup_{x \in \mathcal{M}, y \notin \mathcal{M}} \frac{\mathbb{P}_x(\bar{\tau}_{\mathcal{M} \setminus \{x\}}^\varepsilon < \bar{\tau}_x^\varepsilon)}{\mathbb{P}_y(\bar{\tau}_{\mathcal{M}}^\varepsilon < \bar{\tau}_y^\varepsilon)} = 0. \tag{1}$$

That is, it is much easier to return to $\mathcal{M}$ than to transition between different states in $\mathcal{M}$. The following result, obtained from our perturbative analysis in Appendices B-C, will motivate the distillation scheme described in Section 4.

**Proposition 2.1.** *Any subset $S_\circ = \{x_1, \cdots, x_K\} \subset S$ of cluster representatives $x_k \in C_k$ constitutes a metastable system for $X^\varepsilon$ in the sense of (1) as $M \to \infty$.*

**Meta-chain.** The coarse-grained dynamics of $X^\varepsilon$ over long timescales is captured by its effective **metastable representation** $X_\star^\varepsilon$ (Wicks & Greenwald, 2005; Betz & Le Roux, 2016), which acts as a compression of the full chain by only retaining information on inter-cluster dynamics. This 'meta-chain' is defined on the set of clusters $S_\star = \{C_1, \cdots, C_K\}$ with transition kernel

$$q_\star^\varepsilon(C_\ell | C_k) = \sum_{x \in C_k} \mu_k(x)^2 \mathbb{P}_x(\bar{\tau}_{C_\ell}^\varepsilon < \bar{\tau}_x^\varepsilon), \quad k \neq \ell, \tag{2}$$

and $q_\star^\varepsilon(C_k | C_k)$ such that the conditional probabilities sum to 1. We emphasize that $X_\star^\varepsilon$ faithfully characterizes cluster escape probabilities (see Proposition 4.1) but is *not* a one-to-one copy of the cluster transitions of $X^\varepsilon$, which generally cannot be uniquely defined as a Markov chain. For example, $q_\star^\varepsilon$ is asymptotically reversible and always positive regardless of the actual arrangement of sparse edges (Proposition C.8). To provide further intuition, we state and discuss the following assumption.

**Assumption 4** (uniform escape of $X_\star^\varepsilon$). *For all $k \neq \ell$,*

$$q_\star^\varepsilon(C_\ell | C_k) = \Omega(\varepsilon/M). \tag{3}$$

Equation (3) holds if there exists a sparse edge from $C_k$ to $C_\ell$ (Corollary C.11), but it may well hold even if $C_k, C_\ell$ are not directly connected. For example, if the sparse edges are arranged as a cycle on $S_\star$, escaping $C_k$ implies that all other

**Algorithm 1** Two-stage Pretraining
1: set $\mathbf{W}^{(0)} = \mathbf{0}$, $\eta = O(KM)$,
2: $T_1 = \widetilde{O}(KM^2\varepsilon^{-2})$, $T_2 = \widetilde{O}(KM\varepsilon^{-2})$
3: **for** $t = 1, \cdots, T_1$ **do**
4: $\quad \mathbf{W}^{(t)} = \mathbf{W}^{(t-1)} + \eta\nabla\mathbb{E}_{X_0,X_1}[\log\hat{p}_{\mathbf{W}^{(t-1)}}(X_1|X_0)]$
5: **end for**
6: $w_{ij}^{(T_1)} \leftarrow -\infty$ if $\hat{p}_{ij}^{(T_1)} < c_{\text{thres}}\varepsilon$        (thresholding)
7: **for** $t - T_1 = 1, \cdots, T_2$ **do**
8: $\quad \mathbf{W}^{(t)} = \mathbf{W}^{(t-1)} + \eta\nabla\mathbb{E}_{X_0,X_1}[\log\hat{p}_{\mathbf{W}^{(t-1)}}(X_1|X_0)]$
9: **end for**

clusters $C_\ell$ will be hit before the process returns to $C_k$, and so Assumption 4 is satisfied. Hence Assumption 4 naturally guarantees that it is relatively easy to explore the entire state space from any starting cluster.[1]

# 3. Search Improves the Pretrained Model

## 3.1. Pretraining the Base (World) Model

We equate pretraining the base model with learning the underlying transition kernel $p^\varepsilon$. Indeed, if the context window of an LLM is restricted to the tokens in the previous state, next-token prediction recursively defines a distribution over the following state, and further over reasoning chains of arbitrary length. We encode each state $x \in S$ as a one-hot vector in $\mathbb{R}^{|S|}$ also denoted by $x$ and write $p_{ij}^\varepsilon = p^\varepsilon(e_j|e_i)$. For the model, we consider a simple linear softmax predictor:

$$\hat{p}_{\mathbf{W}}(\cdot|x) = \text{softmax}(\langle\mathbf{W}, x\rangle), \quad \mathbf{W} \in \mathbb{R}^{|S|\times|S|}.$$

The pretraining data consists of random bigram samples $(X_0, X_1)$ where $X_1 \sim p^\varepsilon(\cdot|X_0)$; we allow $X_0$ to be either uniform over $S$ or distributed according to the stationary measure $\pi^\varepsilon$ of $p^\varepsilon$. The latter arises when generating samples $(X_{t-1}, X_t)_{t\geq 1}$ from the observed transitions of the (unbounded) chain $(X_t^\varepsilon)_{t\geq 0}$. The model is trained by gradient descent with cross-entropy loss, with an intermediate thresholding step to mask out edges determined to not lie in $E$. See Algorithm 1 for details and Theorem D.1 for the full statement.

**Theorem 3.1** (convergence of pretraining). *Let $X_0 \sim$ Unif$(S)$ or $X_0 \sim \pi^\varepsilon$ and $X_1 \sim p^\varepsilon(\cdot|X_0)$ be random samples from $X^\varepsilon$. Then for the gradient descent iterates $\mathbf{W}^{(t)}$ from Algorithm 1 w.r.t. cross-entropy loss*

$$L_{\text{pre}}(\mathbf{W}) = \mathbb{E}_{X_0,X_1}[-\log\hat{p}_{\mathbf{W}}(X_1|X_0)],$$

*the learned transition probabilities $\hat{p}_{ij}^{(T)} = \hat{p}_{\mathbf{W}^{(T)}}(e_j|e_i)$ converge with error $\sup_{i,j}|\hat{p}_{ij}^{(T)} - p_{ij}^\varepsilon| = O(\sqrt{KM^2/T})$*

[1] On the other hand, if the meta-chain has poorly connected regions, then $X_\star^\varepsilon$ itself is amenable to metastability analysis, leading to a hierarchy of metastable representations at increasingly faster timescales (Wicks & Greenwald, 2005).

*before thresholding. Moreover, after thresholding at time $T_1 = \widetilde{O}(KM^2\varepsilon^{-2})$, the error converges as $\exp(-\Omega(\varepsilon^2 T))$. Hence after $T_2 = \widetilde{O}(KM\varepsilon^{-2})$ additional steps, the output of Algorithm 1 has error $\exp(-\Omega(|S|))$.*

Thus the base model $\hat{p}$ learns the underlying graph $E$ and all transition probabilities with exponentially small error. Under mild regularity conditions, all assumptions can be verified for $\hat{p}$ (see Propositions B.8 and C.8); to simplify the discussion, we henceforth assume the base model is exact, $\hat{p} = p^\varepsilon$. We remark that while the time to converge is quite long compared to the search, RL and distillation methods studied later, this is natural as pretraining is done on much longer timescales compared to test-time compute.

## 3.2. Learning Sparse Rewards via Search

Having learned the underlying probabilities $p^\varepsilon$, the base model now performs CoT reasoning by generating each step of the chain $(X_t^\varepsilon)_{t\geq 0}$ in sequence starting from $X_0^\varepsilon = X_{\text{in}}$. Since the reasoner has no prior knowledge of which steps it must take to progress towards $X_{\text{out}}$, on average it will spend a long time trapped in each cluster before chancing upon a sparse edge (new idea) and moving to a new cluster. From our quantitative dynamical analysis, we are able to obtain a nearly tight characterization of the average hitting time.

**Theorem 3.2** (expected hitting time). *Under Assumptions 1-4, it holds for all $\varepsilon \leq \varepsilon_{\max} := \Theta(M^{-1}(\log M)^{-4})$ that*

$$\mathbb{E}_{(X_{\text{in}},X_{\text{out}})\sim\mathcal{D}}[\mathbb{E}_{X_{\text{in}}}[\tau_{X_{\text{out}}}^\varepsilon]] = \widetilde{\Theta}\left(\frac{KM}{\varepsilon}\right).$$

Intuitively, since each cluster is rapidly mixing, the chain will spend roughly $\Theta(1/M)$ of the time in states with outbound edges, from where it escapes with probability $\Theta(\varepsilon)$. Such rare events are distributed approximately exponentially, and must be repeated $\Theta(K)$ times to reach the cluster containing $X_{\text{out}}$, where the chain will mix fast and likely hit $X_{\text{out}}$.

This result also illustrates a simple method to improve the hitting time: modifying the underlying probabilities to increase the denominator $\varepsilon$. This corresponds to guiding CoT or fine-tuning the base model so that (correct) new, difficult reasoning steps are generated more often, ensuring a more efficient exploration of the solution space. However, this cannot be done by simply increasing the likelihood of low-probability edges, as there may be many low-probability edges within clusters as well; we want to only boost the generation of sparse edges to preserve the capabilities of the pretrained model. Indeed, we demonstrate in Theorem 5.3 that any updates based on local information is not enough to improve reasoning ability in a precise sense.

Instead, we run a simple *tree search* protocol to identify

**Algorithm 2** Sparse Edge Search

**Require:** pretrained model $\hat{p}_{\mathbf{W}}$
1: set $R = \Theta(K \log K)$, $N = \Theta(\log K)$,
$T_0 = \Theta(M(\log M)^2)$, $T_{\max} = \Theta(M/\varepsilon)$, $\mathcal{M}_s = \varnothing$
2: **for** $r = 1, \cdots, R$ **do**
3:    set $\hat{C}, \hat{C}^n, \hat{E} = \varnothing, A = [N]$
4:    sample $X_0 \sim \mathrm{Unif}(S)$ or $X_0 \sim \pi^\varepsilon$
5:    **for** $t = 1, \cdots, T_{\max}$ **do**
6:      **for** $n \in A$ **do**
7:        generate $X_t^{n,\varepsilon} \sim \hat{p}_{\mathbf{W}}(\cdot | X_{t-1}^{n,\varepsilon})$
8:        **if** $t \leq T_0$ **then**      (cluster search)
9:           $\hat{C}^n \leftarrow \hat{C}^n \cup \{X_t^{n,\varepsilon}\}$
10:       **else if** $t > T_0, X_t^{n,\varepsilon} \notin \hat{C}^n$ **then** (edge search)
11:          $\hat{E} \leftarrow \hat{E} \cup \{(X_{t-1}^{n,\varepsilon}, X_t^{n,\varepsilon})\}$
12:          $A \leftarrow A \setminus \{n\}$
13:       **end if**
14:      **end for**
15:      **if** $t = T_0$ **then**
16:        $\hat{C} = \cap_{n=1}^N \hat{C}^n$
17:      **end if**
18:    **end for**
19:    run Algorithm 3 with $\hat{p}_{\mathbf{W}}, \hat{E}$    (if RL mode)
20:    $\mathcal{M}_s \leftarrow \mathcal{M}_s \cup \hat{E}$       (if PRM mode)
21: **end for**
22: return $\mathcal{M}_s$

---

**Algorithm 3** PPO-Clip

**Require:** pretrained model $\hat{p}_{\mathbf{W}}$, subset of edges $\hat{E}$
1: set $\mathbf{W}^{(0)} = \mathbf{W}$, $T_{\mathrm{PPO}} = \Theta(\log \varepsilon_{\max}/\varepsilon)$, $\alpha = \Theta(KM)$
2: advantage function $\hat{A}(x,y) = 1_{\{(x,y) \in \hat{E}\}}$
3: **for** $t = 1, \cdots, T_{\mathrm{PPO}}$ **do**
4:    $\mathbf{W}^{(t)} = \mathbf{W}^{(t-1)} + \alpha \, \mathrm{sgn}(\nabla L_{\mathrm{PPO}}(\mathbf{W}^{(t-1)}; \hat{A}))$
5: **end for**

---

sparse edges, detailed in Algorithm 2. The method consists of randomly sampling a state $X_0$ and rolling out $N$ random walks in parallel to construct an estimate $\hat{C}$ of the cluster containing $X_0$ for time $T_0$. After the cluster has been sufficiently explored we continue to simulate each walk until a transition outside $\hat{C}$ is detected, at which point the edge is marked as a sparse edge (added to $\hat{E}$) and the path is terminated. This continues until until all paths are terminated or a time horizon $T_{\max}$ is reached. Since we are not receiving signals from an external oracle but rather recording rare transitions, this is similar to intrinsic rewards such as curiosity or exploration bonuses (Burda et al., 2018; 2019).

We consider two versions of this process, **PRM mode** and **RL mode**, depending on whether the information gained from search is collected into an external reward model or used to fine-tune the base model. The benefits of both methods for reasoning is discussed in the next subsection.

### 3.3. Improving the Base Model via RL

**PRM mode** keeps an external process reward 'model' (PRM) throughout the search process, which is simply the set $\mathcal{M}_s$ which collects the estimated sparse edges over multiple iterations of the outer loop to reconstruct $E_s$. We prove that the PRM is strongly consistent:

**Proposition 3.3.** *PRM mode of Algorithm 2 returns $\mathcal{M}_s = E_s$ with probability $1 - \widetilde{O}(1/K)$.*

Then by increasing the likelihood of transitions $(x,y) \in \mathcal{M}_s$ when the current state is $x$ by a factor of $\varepsilon_{\max}/\varepsilon$, the PRM can guide CoT to follow $p^{\varepsilon_{\max}}$ rather than $p^\varepsilon$.
It is immediate from Theorem 3.2 that the expected hitting time decreases from $\widetilde{\Theta}(KM/\varepsilon)$ to $\widetilde{\Theta}(KM/\varepsilon_{\max})$. Moreover, the time complexity of Algorithm 2 is $RT_{\max} = \widetilde{O}(KM/\varepsilon)$, which is equal to the time to solve a *single* instance $(X_{\mathrm{in}}, X_{\mathrm{out}})$ *without* search, and the memory requirement is only $O(M + K)$. This demonstrates the effectiveness of utilizing search to guide CoT generation.

However, it is often desirable to use the information gained during search to directly fine-tune the pretrained model, so that maintaining an independent PRM is not necessary. **RL mode** performs online RL updates to $\hat{p}_{\mathbf{W}}$ at each iteration of Algorithm 2; while many policy gradient methods can be applied, we analyze the popular proximal policy optimization algorithm (PPO-Clip, Schulman et al., 2017). Based on the estimate $\hat{E}$ of sparse edges originating from the initialized cluster, we define the advantage function as $\hat{A}(x,y) = 1$ if $(x,y) \in \hat{E}$ and 0 otherwise. Similarly to pretraining, samples are generated as $X_0 \sim \mathrm{Unif}(S)$ or $\pi^\varepsilon$ and $X_1 \sim p^\varepsilon(\cdot | X_0)$. The objective of PPO-Clip to be maximized is (OpenAI, 2018)

$$L_{\mathrm{PPO}}(\mathbf{W}; \hat{A})$$
$$= \mathbb{E}_{X_0, X_1} \left[ \min \left\{ \frac{\hat{p}_{\mathbf{W}}(X_1 | X_0)}{p^\varepsilon(X_1 | X_0)}, c_{\mathrm{clip}} \right\} \hat{A}(X_0, X_1) \right].$$

The old policy is fixed to $p^\varepsilon$ during Algorithm 2. We use sign gradient ascent for simplicity of analysis (ordinary gradient ascent also guarantees convergence as long as $\varepsilon \geq \varepsilon_{\max}^2$).

**Proposition 3.4** (convergence of PPO-Clip). *By running RL mode of Algorithm 2 with PPO-Clip for a suitable $c_{\mathrm{clip}}$, the base model $p^\varepsilon$ is modified to $p^{\varepsilon'}$ where $\varepsilon' = (1 - o(1))\varepsilon_{\max}$ with probability $1 - \widetilde{O}(1/K)$.*

The additional time complexity of running PPO-Clip is $\widetilde{O}(K \log(\varepsilon_{\max}/\varepsilon))$ which is small compared to the pretraining time or search process. In particular, it again follows from Theorem 3.2 that the expected hitting time is improved by the factor $\varepsilon_{\max}/\varepsilon$. At the same time, the *magnitude* (total variation) of change to the pretrained model is negligible:

$$\sup_{x \in S} \| p^\varepsilon(\cdot | x) - p^{\varepsilon_{\max}}(\cdot | x) \|_{\mathrm{TV}} \leq o(1/M),$$

**Algorithm 4** Meta-chain Distillation

---

1: set $S_\circ = \varnothing$, $\mathbf{Z}^{(0)} = \mathbf{0}$, $\iota(x) = 0$ for all $x \in S$,
2: $T_{\text{dist}} = O(M^2(\log K)^2\varepsilon^{-2})$, $T_{\text{thres}} = \widetilde{O}(M\varepsilon^{-1})$
3: $\eta = \Theta(K)$, $\beta = \Theta(\log(M/\varepsilon))$
4: **while** $\iota^{-1}(0) \neq \varnothing$ **do**         (cluster labeling)
5:     draw $X_0 \in \iota^{-1}(0)$
6:     $S_\circ \leftarrow S_\circ \cup \{X_0\}$, $\iota(X_0) \leftarrow X_0$
7:     **for** $t = 1, \cdots, T_0$ **do**
8:        generate $X_t^\varepsilon \sim p^\varepsilon(\cdot|X_{t-1}^\varepsilon)$
9:        $\iota(X_t^\varepsilon) \leftarrow X_0$
10:    **end for**
11: **end while**
12: **for** $t = 1, 2, \cdots$ **do**           (data collection)
13:     **if** $X_t^\varepsilon \in S_\circ$ **then**
14:        $Y_0^{(t)}, Y_1^{(t_{\text{prev}})} \leftarrow X_t^\varepsilon$
15:        $t_{\text{prev}} \leftarrow t$
16:     **else**
17:        $Y_0^{(t)}, Y_1^{(t)} \leftarrow \iota(X_t^\varepsilon)$
18:     **end if**
19: **end for**
20: **for** $t = 1, \cdots, T_{\text{dist}}$ **do**       (distillation)
21:     $\mathbf{Z}^{(t)} = \mathbf{Z}^{(t-1)} - \eta \nabla \mathbb{E}_{Y_0, Y_1}[-\log \hat{p}_{\mathbf{Z}^{(t-1)}}(Y_1|Y_0)]$
22:     **if** $t = T_{\text{thres}}$ **then**
23:        $z_{k\ell}^{(T_{\text{thres}})} \leftarrow -\infty$ if $\hat{q}_{k\ell}^{(T_{\text{thres}})} < c_{\text{thres}}\varepsilon/M$
24:     **end if**
25: **end for**
26: $z_{k\ell}^+ \leftarrow z_{k\ell}^{(T_{\text{dist}})} + \beta$ for $\ell \neq k$      (time rescaling)
27: **return** $\mathbf{Z}^+$

---

so the original capabilities of the base model are generally preserved. Hence RL is also extremely efficient for fine-tuning the pretrained model to improve CoT.

# 4. Distillation to a Smaller Model

A prominent innovation in the LLM development pipeline is to distill CoT of a powerful model into a smaller, more efficient model. This approach has been shown to significantly enhance reasoning ability, especially compared to directly training the smaller model with RL (Shridhar et al., 2022; Hsieh et al., 2023; Gandhi et al., 2024; Guo et al., 2025). In this section, we showcase an explicit distillation scheme for our CoT model that efficiently generates the hard reasoning steps to solve any task while faithfully capturing the metastable dynamics of the original system.

## 4.1. Distilling Cluster Transitions

The metastable chain $q_\star^\varepsilon$ (Section 2) provides a natural notion of compression for the nearly reducible system $X^\varepsilon$ by collapsing each cluster into a single state. For many down-stream tasks (including the logic task studied in Section 5) it may be satisfactory to retrieve only the hard reasoning steps

connecting the clusters containing $X_{\text{in}}, X_{\text{out}}$. In particular, if the goal is to extract only the *connectivity* of $S_\star$, it suffices to take the sparse edge estimate $\mathcal{M}_s$ of Algorithm 2 and perform a uniform random walk to find a path between any two clusters. However, we want the distilled model to also preserve the underlying dynamics of the original chain as best as possible. To this end, we implement the following process, detailed in Algorithm 4.

We first choose a set $S_\circ = \{x_1, \cdots, x_K\}$ of representatives $x_k$ of $C_k$ and assign each state to its representative via the map $\iota: S \to S_\circ$; this can be done by exploring each cluster similarly to the first $T_0$ steps of the search process.

**Data collection.** The data for distillation is collected by continually running CoT and recording the frequency of transitions (or non-transitions) between $S_\circ$. The yields one datum per CoT step, and can also be implemented in parallel for an arbitrary number of independent chains.

(1) If $X_t^\varepsilon \in S_\circ$ and the previous return to $S_\circ$ was $X_{t_{\text{prev}}}^\varepsilon$ then add $(X_{t_{\text{prev}}}^\varepsilon, X_t^\varepsilon)$ to $D_{\text{dist}}$.

(2) If $X_t^\varepsilon \notin S_\circ$ (no transition) add $(\iota(X_t^\varepsilon), \iota(X_t^\varepsilon))$ to $D_{\text{dist}}$.

This requires only $O(K^2)$ memory for frequency counts; no cache for $X^\varepsilon$ is needed. The cluster labels $\iota$ and parameters $\mathbf{Z}$ require $O(KM)$ and $O(K^2)$ memory, respectively. We suppose the process is run for arbitrarily long time so that we have access to the population distribution of $D_{\text{dist}}$. We then one-hot embed $S_\circ$ in $\mathbb{R}^K$ and use the collected data pairs to train a softmax model $\hat{q}_{\mathbf{Z}}(\cdot|x) = \text{softmax}(\langle \mathbf{Z}, x\rangle)$, $\mathbf{Z} \in \mathbb{R}^{K \times K}$ similarly to pretraining. Finally, we rescale time so that the non-diagonal entries sum to $\Theta(1)$, reducing redundant within-cluster transitions.

**Equivalence with meta-chain.** The data $(Y_0, Y_1) \sim D_{\text{dist}}$ has been constructed so that the distilled model learns the following kernel $q_\circ^\varepsilon$ on $S_\circ$: $Y_0 \sim \pi^\varepsilon$, $Y_1 \sim q_\circ^\varepsilon(\cdot|Y_0)$ where

$$q_\circ^\varepsilon(x_\ell|x_k) := \pi_k^\varepsilon(x_k)\mathbb{P}_{x_k}(X_{\bar{\tau}_{S_\circ}^\varepsilon}^\varepsilon = x_\ell), \quad k \neq \ell,$$
$$q_\circ^\varepsilon(x_k|x_k) := 1 - \sum_{\ell \neq k} q_\circ^\varepsilon(x_\ell|x_k).$$

This kernel is a lazy version of the process obtained from $X^\varepsilon$ by deleting all transitions to states outside $S_\circ$, with an additional time rescaling according to the stationary probability $\pi_k^\varepsilon(x_k)$. This is slightly different from the construction given in Betz & Le Roux (2016), as we do not presume access to the stationary distribution of the unperturbed chain $p_k^0$ and must sample directly from $\pi^\varepsilon$. Moreover, $q_\circ^\varepsilon$ is dependent on the choice of representatives and thus different from the 'canonical' meta-chain $q_\star^\varepsilon$ in general. Nonetheless, $q_\circ^\varepsilon$ is faithful to the meta-chain in a rigorous sense:

**Proposition 4.1.** *Denote the return time of $q_\circ^\varepsilon$ to $x_k$ as $\bar{\tau}_{\circ, x_k}^\varepsilon$.*

*For all $k, \ell \in [K]$ with $k \neq \ell$, it holds that*

$$\frac{\mathbb{P}_{x_k}(\bar{\tau}^\varepsilon_{\circ,x_\ell} < \bar{\tau}^\varepsilon_{\circ,x_k})}{q^\varepsilon_\star(C_\ell | C_k)} = 1 + o_M(1).$$

That is, the escape probabilities of $q^\varepsilon_\circ$ converge to $q^\varepsilon_\star$ with uniformly vanishing relative error. This property is desirable as it shows the eventual likelihood of escaping to each cluster (i.e., reaching a certain idea) is consistent across different choices of $S_\circ$.

## 4.2. CoT of Distilled Model

To analyze the utility of the trained model $\hat{p}_{\mathbf{Z}^+}$, we make the additional assumption:

**Assumption 5** (inbound sparse edges). *All sparse edges leading to each cluster $C_k$ terminate at a fixed point $x_k$. For any sparse edge $(x', x_\ell)$ from $C_k$, there exists a path from $x_k$ to $x'$ in $C_k$ of probability bounded below.*

Then we may specify $S_\circ$ as the set of the points $x_k$. This ensures that representatives will not be skipped; otherwise, a CoT passing through $C_k, C_\ell, C_n$ in succession may miss $x_\ell$ and record the wrong transition $(x_k, x_n)$ (although the likelihood of this is $o(1)$ regardless).

Now, as with pretraining, the distilled model will converge to $q^\varepsilon_\circ$ when trained with cross-entropy loss on $D_{\text{dist}}$.

**Proposition 4.2** (convergence of distillation). *For the gradient descent iterates $\mathbf{Z}^{(t)}$ from Algorithm 4, the learned probabilities converge to $q^\varepsilon_\circ$ after $T_{\text{dist}} = \widetilde{O}(M^2 \varepsilon^{-2})$ as*

$$\sup_{k,\ell} |\hat{q}_{\mathbf{Z}^{(T_{\text{dist}})}}(x_\ell | x_k) - q^\varepsilon_\circ(x_\ell | x_k)| = K^{-\omega(1)}.$$

We point out the time to convergence is much faster than pretraining time $\widetilde{O}(KM^2 \varepsilon^{-2})$ (Theorem 3.1), and also more computationally efficient since we are training a size $K^2$ model rather than size $(KM)^2$.

Finally, after time rescaling, the model $q_{\mathbf{Z}^+}$ is capable of efficiently finding a path from the cluster containing $X_{\text{in}}$ to the cluster containing $X_{\text{out}}$, with hitting time linear in $|S_\circ|$ and *independent of the difficulty parameter $\varepsilon$.*

**Theorem 4.3** (hitting time of distilled CoT). *For all $k \neq \ell$, $\hat{q}_{\mathbf{Z}^+}(x_\ell | x_k) = \Theta(1)$ if there exists a sparse edge from $C_k$ to $C_\ell$ or $0$ if not. Moreover, the hitting time $\tau^+_{x_\ell}$ of $x_\ell \in S_\circ$ by $\hat{q}_{\mathbf{Z}^+}$ satisfies $\mathbb{E}_{x_k}[\tau^+_{x_\ell}] = O(K)$.*

The returned sequence of clusters $C_{0:T}$ indicate the existence of a path from $X_{\text{in}}$ to $X_{\text{out}}$ passing through precisely these clusters in order. Once $C_{0:T}$ is determined, a weaker reasoning agent (e.g., the base model $p^\varepsilon$) may also efficiently resolve the fine-grained dynamics within each cluster.

## 5. Logical Reasoning is Hard without Search

### 5.1. Logical Reasoning Task

In this section, we further investigate the benefits of search for reasoning by adding a quantitative 'logic task' on top of the path-finding task. This provides two benefits. First, having a numerical answer allows us to evaluate the hardness of the task from a learning-theoretic perspective, separate from the previously obtained hitting time bounds. Second, by having the answer depend only on the sparse edges along a path, the reasoner is required to estimate which edges are sparse – in other words, understand which reasoning steps are actually important – in order to solve the task. Taking a proof problem for example, we expect an LLM with strong reasoning capability to not only *generate* a plausible solution via next-token prediction but also *understand* its own proof, so that it can correctly answer logical questions such as "what are the key ideas of this proof?" or "what happens if we replace step X with Y?" We attempt to formalize this notion using group actions (Definition E.1).

**Logical actions.** Let $(G, \circ)$ be a finite group with identity $e_G$. The *logical value* (or simply *logic*) of a reasoning chain is an element of an abstract space $\mathcal{R}$ equipped with a $G$-action $r \mapsto g \cdot r$. Each edge $e \in E$ is assigned a *logical action* $\alpha(e) \in G$ which acts on the current logic when the edge is selected. To focus on learning hard steps, we assume that the logical action of edges not in $E_s$ are trivial, $\alpha|_{E_s^c} := e_G$. Let $\psi : S \to \mathcal{R}$ be an arbitrary embedding map. For a valid path $X_{0:T} \subseteq S$, we define the corresponding logic sequence $r_{0:T} \subseteq \mathcal{R}$ as

$$r_0 = \psi(X_0), \quad r_t = \alpha(X_{t-1}, X_t) \cdot r_{t-1}.$$

For example, if each state is a Boolean expression being manipulated according to certain rules, $\mathcal{R} = G = \mathbb{Z}_2$ could be used to encode the evaluation of the current expression by switching between $1$ (True) and $0$ (False) depending on the effect of each manipulation. $G$ could also be taken to be a space of functions with the evaluation action $g \cdot r = g(r)$, so that the logic computes a repeated composition of functions. When the chain terminates, the final logic $r_T =: r(X_{0:T})$ is returned. Note that logical values are not unique to states and $r_T$ depends on the entire path $X_{0:T}$.

**Logic Task.** Given $(X_{\text{in}}, X_{\text{out}}) \sim \mathcal{D}$, the goal is to output both a valid path $X_{0:T}$ from $X_{\text{in}}$ to $X_{\text{out}}$ and its logical value $r(X_{0:T})$. Since any path can be made simple by deleting loops, here we require valid paths to be simple.

To establish a rigorous distinction between the use of a search algorithm and lack thereof, we consider models consisting of a pretrained base model or *linguistic* component $\mathcal{M}_p$, responsible for learning $p$ and generating a valid CoT,

and a *reasoning* component $f_\theta$, which predicts the answer $r(X_{0:T})$ based on (limited) information from $\mathcal{M}_p$. As in Section 3, we suppose $\mathcal{M}_p$ has perfectly learned the kernel $p$ and can output arbitrary valid paths $\mathcal{M}_p(X_{\text{in}}, X_{\text{out}})$, solving the first part of the task. Here we do not consider the time complexity of running $\mathcal{M}_p$, which (as we have seen in Theorem 3.2) can be quite long without a search-and-improvement protocol. Thus the main task of the reasoner is to execute logical computations along a generated CoT.

In this section, we assume a stronger *uniform* lower bound in Assumption 3 on the minimum number of hard steps; otherwise, querying a single sparse edge $(X_{\text{in}}, X_{\text{out}}) \in E_s(p)$ could immediately reveal its (nontrivial) action.

**Assumption 3'.** For any $(X_{\text{in}}, X_{\text{out}}) \sim \mathcal{D}$ and any valid path $X_{0:T}$ with $X_0 = X_{\text{in}}$ and $X_T = X_{\text{out}}$, it holds that $\min |X_{0:T} \cap E_s| = \Omega(K)$.

We remark that this condition can be weakened to $\Omega(\log K)$ if $M \geq \Omega(K)$, in which case our results hold with $e^{-\Omega(K)}$ replaced by $K^{-\omega(1)}$.

**Concept class.** Define $\mathcal{P}$ the set of transition kernels on $S$ satisfying Assumptions 1, 2 and denote the sparse edge set of $p \in \mathcal{P}$ as $E_s(p)$. The logical action $\alpha$ can be seen as generated by sampling $\mathcal{A} : S \times S \to G$ i.i.d. uniformly from $G$, then masking out all edges not in $E^s(p)$ by setting them to $e_G$. Thus $\mathcal{A}$ can be regarded as a variable separate from the target $p$, and the logic is computed recursively as

$$r_{\mathcal{A},p}(X_0) = \psi(X_0), \quad r_{\mathcal{A},p}(X_{0:t}) =$$
$$\begin{cases} \mathcal{A}(X_{t-1}, X_t) \cdot r_{\mathcal{A},p}(X_{0:(t-1)}) & (X_{t-1}, X_t) \in E_s(p) \\ r_{\mathcal{A},p}(X_{0:(t-1)}) & (X_{t-1}, X_t) \notin E_s(p). \end{cases}$$

Finally, the logic $r_{\mathcal{A},p}(X_{0:T})$ is mapped to a scalar output via a classifier $\phi : \mathcal{R} \to \{+1, -1\}$. We assume that $\mathbb{E}_{g \in G}[\phi(g \cdot r)] = 0$ for all $r \in \mathcal{R}$. The concept class is thus

$$\mathcal{H} = \big\{ h_p \in S \times S \times G^{|S| \times |S|} : p \in \mathcal{P},$$
$$h_p(X_{\text{in}}, X_{\text{out}}, \mathcal{A}) = \phi \circ r_{\mathcal{A},p}(\mathcal{M}_p(X_{\text{in}}, X_{\text{out}})) \big\},$$

equipped with inner product $\langle h_p, h_{p'} \rangle_{\mathcal{H}} := \mathbb{E}_{(X_{\text{in}}, X_{\text{out}}) \sim \mathcal{D}, \mathcal{A}}[h_p(X_{\text{in}}, X_{\text{out}}, \mathcal{A}) h_{p'}(X_{\text{in}}, X_{\text{out}}, \mathcal{A})]$.

### 5.2. A Measure of Hardness with Restricted Access

In previous sections, we have seen that pretraining $\mathcal{M}_p = \hat{p}_{\mathbf{W}}$ and running a search or distillation algorithm $f_\theta$ will correctly infer the underlying sparse structure. In this case, computing $r_{\mathcal{A},p}(\mathcal{M}_p(X_{\text{in}}, X_{\text{out}}))$ is trivial by concatenating actions along the identified sparse edges. In contrast, we now restrict the reasoning component's access to $p$ by only allowing certain queries to $\mathcal{M}_p$. This makes it difficult to infer the sparse structure and true logical actions.

To understand learning with this additional (restricted) information, we propose the following generalization of the statistical query dimension (Kearns, 1998; Feldman, 2017).

**Definition 5.1** (SDA: SQDIM with access). Let $\mathcal{P}$ be the set of ground truths and $\mathcal{H} = \{h_p : \mathcal{X} \to \{\pm 1\} \mid p \in \mathcal{P}\}$ the associated concept class with inner product $\langle \cdot, \cdot \rangle_{\mathcal{H}}$. Let $\mathcal{I}_p$ be any value or any function on $\mathcal{X}$ depending on $p$. Then the *statistical query dimension of $\mathcal{P}$ with access to $\mathcal{I}$* and tolerance $\tau$ is defined as

$$\text{SDA}_\tau(\mathcal{P}; \mathcal{I}) := \sup\{|\mathcal{P}'| : \mathcal{P}' \subseteq \mathcal{P},$$
$$|\langle h_{p_1}, h_{p_2} \rangle_{\mathcal{H}}| \leq \tau, \mathcal{I}_{p_1} = \mathcal{I}_{p_2} \, \forall p_1 \neq p_2 \in \mathcal{P}'\}.$$

In this section, we consider $\tau = 0$ and omit its notation. Extending classical analyses (e.g., Shalev-Shwartz et al., 2017; Shamir, 2018), we prove a general limitation for gradient-based learning when additional information $\mathcal{I}_p$ is provided.

**Theorem 5.2** (SQ learning with additional information). *Let $f_\theta$ be any parametric model of the form*

$$x \mapsto f_\theta(x, \mathcal{I}_p(x)).$$

*Let the loss function be $L(\theta; p) := \|h_p - f_\theta\|_{\mathcal{H}}^2$ and set $\delta := (4\|\nabla f_\theta\|_{\mathcal{H}}^2 / \text{SDA}(\mathcal{P}, \mathcal{I}))^{1/3}$. Then choosing $p$ randomly from a subset of $\mathcal{P}$, any iterative algorithm $A(\theta)$ that makes at most $n$ queries to the $\delta$-corrupted gradient oracle $\nabla L$ has expected loss*

$$\mathbb{E}_p[L(A(\theta); p)] \geq 1 - \text{SDA}(\mathcal{P}, \mathcal{I})^{-1}$$

*with probability at least $1 - n\delta$.*

Note that we only consider the squared loss in our formulation for simplicity. While squared loss only answers correlational queries, CSQ-learnability is equivalent to SQ-learnability for Boolean concepts (Bshouty & Feldman, 2001).

### 5.3. Results on Hardness of Logical Task

We consider four types of access to the pretrained model. Note that a *local neighborhood* of a subset $S' \subset S$ in the weighted directed graph defined by $p$ is defined as the subgraph consisting of states reachable with a bounded number of steps from any state in $S'$.

(1) **No pretraining**, $\mathcal{I}_p \equiv \varnothing$: the learner $f_\theta(X_{\text{in}}, X_{\text{out}}, \mathcal{A})$ has not been pretrained and does not receive any information on $p$.

(2) **Path-only (no search)**, $\mathcal{I} \equiv \mathcal{M}$: the learner is allowed to depend on inputs $X_{\text{in}}, X_{\text{out}}$, and $\mathcal{A}$, and also the generated path $\mathcal{M}_p(X_{\text{in}}, X_{\text{out}})$. That is, the linguistic component (base model) will return a valid CoT for the input at hand, but we cannot simulate different chains from $p$ to execute some search policy or inference algorithm.

(3) **Local search**, $\mathcal{I} \equiv \mathrm{nbd}(\mathcal{M})$: the learner is allowed full access to a local neighborhood of $\mathcal{M}_p(X_{\mathrm{in}}, X_{\mathrm{out}})$ in the graph of $p$, including connectivity information and transition probabilities. For instance, it can flag low-probability edges as more likely to be sparse, or run bounded-length CoT from $X_{\mathrm{in}}$ or $X_{\mathrm{out}}$.

(4) **Full search**, $\mathcal{I} \equiv \mathcal{P}$: the learner is given full access to the entire graph of $p$ at all times. In this case, Algorithm 2 or 4 can be used to infer $E_s(p)$ and generate CoT efficiently, and also perform the desired computation $h_p$.

Our main negative result states that (1)-(3) *cannot* solve the logic task with polynomial compute, and thus global search is necessary:

**Theorem 5.3.** $\mathrm{SDA}(\mathcal{P}; \mathcal{P}) = 1$ *and*

$$\mathrm{SDA}(\mathcal{P}; \varnothing) \geq \mathrm{SDA}(\mathcal{P}; \mathcal{M}) \geq \mathrm{SDA}(\mathcal{P}; \mathrm{nbd}(\mathcal{M})) \geq e^{\Omega(K)}.$$

**Remark 5.4.** The necessity of global information for certain learning problems (*globality barrier*) has been conjectured in Abbe et al. (2024), where the hardness of a 'cycle task' is proved. These results are also closely related to classical SQ-hard problems such as subset parity. The precise relationship between SDA, globality and learning is still open.

**Remark 5.5.** While it suffices to lower bound the strictest term $\mathrm{SDA}(\mathcal{P}; \mathrm{nbd}(\mathcal{M}))$, we exhibit different constructions for each of the three dimensions as they offer increasing levels of generality. In particular, $\mathrm{SDA}(\mathcal{P}; \varnothing)$ can be realized by $\mathcal{P}' \subset \mathcal{P}$ containing any prescribed $p \in \mathcal{P}$ and for any $\mathcal{D}$. Moreover, the difficulty is solely due to the logical part of the task; without pretraining, the reasoner will take exponentially many guesses to even produce a valid path.

**Corollary 5.6** (hardness without global search). *Suppose* $f_\theta(\mathrm{nbd}(\mathcal{M}_p(X_{\mathrm{in}}, X_{\mathrm{out}})), \mathcal{A})$ *is any parametric model with polynomially bounded gradients, that can freely search a local neighborhood of the generated CoT. Then any iterative algorithm $A(\theta)$ that makes at most polynomial queries to the $e^{-\Omega(K)}$-corrupted gradient oracle $\nabla L$ satisfies*

$$\mathbb{E}_p[L(A(\theta); p)] \geq 1 - e^{-\Omega(K)},$$

*with probability $1 - e^{-\Omega(K)}$ for $M$ sufficiently large.*

Hence $\mathcal{H}$ cannot be even weakly learned in polynomial time if search is not long enough. The key intuition is that if the graph is locally isomorphic, local search cannot distinguish between sparse inter-cluster edges and low-probability but within-cluster edges as it cannot explore the whole cluster. This demonstrates the importance of spending sufficient inference-time compute for improving reasoning ability.

## 6. Conclusion

We introduced a metastable Markov framework for modeling CoT reasoning, revealing the benefits of inference-time search, RL, and distillation. We showed that search can improve reasoning by identifying critical sparse transitions (hard steps), which can then be leveraged to fine-tune the pretrained model via RL or distilled into a more efficient representation, improving hitting times for path generation. We further established learning-theoretic limits on reasoning with restricted information and showed that logical reasoning tasks become intractable without global search.

**Future directions.** We have studied a simple curiosity-based unsupervised reward model; it would be interesting to see how a more complex search process could be guided with outcome rewards. Our framework could also be used to study other inference-time methods such as CoT revision (e.g., backtracking to better locate sparse edges), as well as iterative finetuning of the pretrained model, and explore scaling laws for inference time compute.

## Acknowledgements

This research is supported by the National Research Foundation, Singapore and the Ministry of Digital Development and Information under the AI Visiting Professorship Programme (award number AIVP-2024-004). Any opinions, findings and conclusions or recommendations expressed in this material are those of the author(s) and do not reflect the views of National Research Foundation, Singapore and the Ministry of Digital Development and Information.

The authors thank Jimmy Ba and Yuexiang Zhai for helpful discussions. JK was partially supported by JST CREST (JPMJCR2015). JDL acknowledges support of the NSF CCF 2002272, NSF IIS 2107304, and NSF CAREER Award 2144994. TS was partially supported by JSPS KAKENHI (24K02905, 20H00576) and JST CREST (JPMJCR2115). This research is unrelated to DW's work at xAI.

## Impact Statement

This paper presents work whose goal is to advance the field of Machine Learning. There are many potential societal consequences of our work, none which we feel must be specifically highlighted here.

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

# A. Additional Related Works

**Theoretical Analysis of CoT.** Some theoretical works have focused on the expressivity of CoT (Feng et al., 2023a; Merrill & Sabharwal, 2023; Chiang et al., 2023; Li et al., 2024b), analysis of optimization and estimation ability (Li et al., 2024a; Hu et al., 2024; Kim & Suzuki, 2024), or in-context learning ability (Li et al., 2023; Bhattamishra et al., 2024). More closely related to our paper, Sanford et al. (2024b;a); Abbe et al. (2024) study the algorithmic reasoning capabilities of CoT or scratchpad transformers for certain computational or graph-based tasks. Also, Nichani et al. (2024) analyze how simple transformer models learn latent causal structure within the data.

**LLMs as Markov processes.** Zekri et al. (2024) investigate the equivalence between autoregressive models and general length Markov chains.Makkuva et al. (2024); Edelman et al. (2024) model sequential data as a Markov chain and analyze the properties of a single-layer transformer. Ildiz et al. (2024) establish a link between self-attention and context-conditioned Markov models. Such works generally focus on interpreting next-token prediction of a specific architecture, and do not consider the abstraction to CoT reasoning.

**Metastable Markov chains.** The literature on metastable Markov processes is vast (e.g., Madras & Randall, 2001; Bovier et al., 2002; Beltrán & Landim, 2011; Landim, 2018). Here we only mention the results most relevant to our theory. In particular, as reversibility is unrealistic to presume for language or reasoning models, we generally restrict our attention to works on nonreversible processes. Fritzsche et al. (2008); Jacobi (2010); Tifenbach (2011); Fackeldey et al. (2018) study various spectral methods to identify metastable states of Markov chains. Landim (2012); Cirillo et al. (2014); Fernandez et al. (2014; 2016); Bianchi & Gaudillière (2016) analyze critical configurations and escape times for metastable dynamics, while Landim & Xu (2015) propose a recursive procedure for model reduction. Most relevant to our work, Wicks & Greenwald (2005); Betz & Le Roux (2016) give a complete hierarchical characterization of the effective dynamics of perturbed chains but only in the asymptotic limit; building on their results, we develop a new quantitative perturbation analysis throughout Appendix C.

# B. Preliminaries

## B.1. Pseudo-Spectral Gap and Mixing

By taking Assumption 2 and multiplying $\varepsilon$ by a constant if necessary, we assume that $c\varepsilon \leq p^\varepsilon(y|x) \leq \varepsilon$ for some $c > 0$ and all $(x, y) \in E_s$ throughout the appendix.

**Definition B.1** (mixing time). For a time-homogeneous ergodic Markov chain $X = (X_t)_{t \geq 0}$ on a finite state space $\Omega$ with transition kernel $p$ and stationary distribution $\pi$, the mixing time $t_{\text{mix}}$ is defined as

$$t_{\text{mix}}(\epsilon) = \min \left\{ t \geq 0 : \forall s \geq t, \ \sup_{x \in \Omega} \|p^s(\cdot|x) - \pi\|_{\text{TV}} \leq \epsilon \right\}.$$

**Definition B.2** (hitting and return times). The $n$th *hitting time* and *return time* of $X^\varepsilon$ to a set $A \subseteq S$ for $n \in \mathbb{N}$ are defined as

$$\tau^\varepsilon_{A,n} = \inf\{t \geq 0 : |\{0 \leq t' \leq t : X^\varepsilon_{t'} \in A\}| = n\},$$
$$\bar{\tau}^\varepsilon_{A,n} = \inf\{t > 0 : |\{0 < t' \leq t : X^\varepsilon_{t'} \in A\}| = n\}.$$

In particular, we write $\tau^\varepsilon_A = \tau^\varepsilon_{A,1}$ and $\bar{\tau}^\varepsilon_A = \bar{\tau}^\varepsilon_{A,1}$. We write $\tau^\varepsilon_x = \tau^\varepsilon_{\{x\}}$, etc. for simplicity.

The chain $X$ is *reversible* if it satisfies the *detailed balance equation*

$$\pi(x)p(y|x) = \pi(y)p(x|y) \quad \forall x, y \in \Omega.$$

While we do not assume reversibility in this paper, it is informative to compare the conditions for rapid mixing. Denote the transition matrix corresponding to $p$ by $\mathbf{P}$ and let the eigenvalues of $\mathbf{P}$ ordered by absolute value be $1 = \lambda_1(\mathbf{P}) \geq |\lambda_2(\mathbf{P})| \geq |\lambda_3(\mathbf{P})| \geq \cdots$. For reversible chains, all eigenvalues are real and the mixing time is closely governed by the (absolute) spectral gap $\gamma(\mathbf{P}) = 1 - |\lambda_2(\mathbf{P})|$ (Levin et al., 2009):

$$\frac{1}{2 \log 2\epsilon} \left( \frac{1}{\gamma(\mathbf{P})} - 1 \right) \leq t_{\text{mix}}(\epsilon) \leq \frac{1}{\gamma(\mathbf{P})} \log \frac{1}{\pi_* \epsilon},$$

where $\pi_* = \min_{x \in \Omega} \pi(x)$ is the minimum stationary probability.

For nonreversible chains, the analogous quantity to $\gamma(\mathbf{P})$ is given by the *pseudo-spectral gap*:

**Definition B.3** (Paulin (2015))**.** The *pseudo-spectral gap* of $\mathbf{P}$ is given as

$$\gamma^\dagger(\mathbf{P}) := \max_{m \in \mathbb{N}} \frac{\gamma((\mathbf{P}^\dagger)^m \mathbf{P}^m)}{m}$$

where $\mathbf{P}^\dagger$ is the time reversal of $\mathbf{P}$, defined as $\mathbf{P}^\dagger_{ij} = \pi_j \mathbf{P}_{ji}/\pi_i$.

When $X$ is reversible, it holds that $\gamma(\mathbf{P}) \leq \gamma^\dagger(\mathbf{P}) \leq 2\gamma(\mathbf{P})$ (Wolfer & Kontorovich, 2022, Lemma 15). Moreover, $\gamma^\dagger(\mathbf{P})$ controls the mixing time similarly to $\gamma(\mathbf{P})$:

**Proposition B.4** (Paulin (2015), Proposition 3.4)**.** *For* $0 < \epsilon < 1$,

$$\frac{1 - 2\epsilon}{\gamma^\dagger(\mathbf{P})} \leq t_{\mathrm{mix}}(\epsilon) \leq \frac{1}{\gamma^\dagger(\mathbf{P})} \left( 1 + 2 \log \frac{1}{2\epsilon} + \log \frac{1}{\pi_*} \right).$$

Denote the maximum row sum norm as $\|\mathbf{A}\|_{1,\infty} = \max_i \sum_j |a_{ij}|$ for $\mathbf{A} = (a_{ij})$.

**Lemma B.5.** *For* $\mathbf{A} \in \mathbb{R}^{m \times m}$ *it holds that* $\|\mathbf{A}\|_2 \leq \sqrt{m}\|\mathbf{A}\|_{1,\infty}$.

*Proof.* For arbitrary $v \in \mathbb{R}^m$ with $\|v\| = 1$,

$$\|\mathbf{A}v\|^2 = \sum_i \left( \sum_j a_{ij} v_j \right)^2 \leq \sum_i \sum_j a_{ij}^2 \leq \sum_i \left( \sum_j |a_{ij}| \right)^2 \leq m\|\mathbf{A}\|_{1,\infty}^2.$$

$\square$

## B.2. Stochastic Complementation

We denote the stochastic block matrix $\mathbf{P}^\varepsilon$ corresponding to the kernel $p^\varepsilon$ and partition $S = \cup_{k=1}^K C_k$ as

$$\mathbf{P}^\varepsilon = \begin{pmatrix} \mathbf{P}^\varepsilon_{11} & \cdots & \mathbf{P}^\varepsilon_{1K} \\ \vdots & \ddots & \vdots \\ \mathbf{P}^\varepsilon_{K1} & \cdots & \mathbf{P}^\varepsilon_{KK} \end{pmatrix}.$$

That is, the probability $p^\varepsilon(y|x)$ is contained in the $(x, y)$ component, and the rows of $\mathbf{P}^\varepsilon$ all sum to 1. The *stochastic complement* of $\mathbf{P}^\varepsilon_{kk}$ is defined as (Meyer, 1989)

$$\mathbf{S}^\varepsilon_{kk} = \mathbf{P}^\varepsilon_{kk} + \mathbf{P}^\varepsilon_{k*}(\mathbf{I} - \mathbf{P}^\varepsilon_k)^{-1}\mathbf{P}^\varepsilon_{*k}$$

where $\mathbf{P}^\varepsilon_{k*}$ is the $k$th block row of $\mathbf{P}^\varepsilon$ with $\mathbf{P}^\varepsilon_{kk}$ removed; $\mathbf{P}^\varepsilon_{*k}$ is the $k$th block column of $\mathbf{P}^\varepsilon$ with $\mathbf{P}^\varepsilon_{kk}$ removed; and $\mathbf{P}^\varepsilon_k$ is the principal block submatrix of $\mathbf{P}^\varepsilon$ with the $k$th row and column removed. When $\varepsilon = 0$, it follows that $\mathbf{P}^0_{ij} = \mathbf{0}$ when $i \neq j$ and $\mathbf{S}^0_{kk} = \mathbf{P}^0_{kk}$ is the transition matrix of $p^0$ restricted to $C_k$.

The following results are fundamental to the theory of stochastic complementation.

**Theorem B.6** (Meyer (1989), Theorem 2.3)**.** *If* $\mathbf{P}^\varepsilon$ *is an irreducible stochastic matrix for* $\varepsilon > 0$, *each stochastic complement* $\mathbf{S}^\varepsilon_{kk}$ *is also an irreducible stochastic matrix. Moreover,* $\mathbf{S}^\varepsilon_{kk}$ *is equal to the transition matrix of the reduced chain* $\tilde{X}^{k,\varepsilon}$ *on* $C_k$,

$$\tilde{X}^{k,\varepsilon}_t := X^\varepsilon_{\tau^\varepsilon_{C_k, t+1}}, \quad t \in \mathbb{N}_0 \tag{4}$$

*obtained from* $X^\varepsilon$ *by deleting transitions to states outside of* $C_k$.

We further denote the transition kernel of $\tilde{X}^{k,\varepsilon}_t$ corresponding to $\mathbf{S}^\varepsilon_{kk}$ as $s^\varepsilon_{kk}$, so that $s^0_{kk} = p^0|_{C_k}$, and its return time to a subset $A \subseteq C_k$ as $\tilde{\tau}^{k,\varepsilon}_A$.

**Lemma B.7** ([Meyer](1989), Theorem 6.1). *Denoting the block diagonal matrix* $\mathbf{S}^\varepsilon = \operatorname{diag} \mathbf{S}^\varepsilon_{kk}$, *it holds for all* $k \in [K]$ *that*

$$\|\mathbf{S}^\varepsilon_{kk} - \mathbf{P}^\varepsilon_{kk}\|_{1,\infty} = \|\mathbf{P}^\varepsilon_{k*}\|_{1,\infty} \quad and \quad \|\mathbf{S}^\varepsilon - \mathbf{P}^\varepsilon\|_{1,\infty} = 2 \max_{k \in [K]} \|\mathbf{P}^\varepsilon_{k*}\|_{1,\infty}.$$

In particular, it immediately follows that

$$\|\mathbf{S}^\varepsilon_{kk} - \mathbf{P}^\varepsilon_{kk}\|_{1,\infty} \le d_{\text{out}}\varepsilon, \quad \|\mathbf{S}^\varepsilon - \mathbf{P}^\varepsilon\|_{1,\infty} \le 2d_{\text{out}}\varepsilon. \tag{5}$$

We now exhibit conditions on the unperturbed matrices $\mathbf{P}^0_{kk}$ which imply the bounds on the spectral gap and stationary distribution of $\mathbf{S}^\varepsilon_{kk}$ in Assumption 1 up to a constant factor. The argument can be repeated to show that Assumption 1 holds for the pretrained model $\hat{p}$ of Theorem 3.1, as the error is exponentially small. This also implies that Assumption 4 is robust to perturbation via Proposition C.8.

**Proposition B.8.** *Suppose that each* $p^0|_{C_k}$ *is reversible with spectral gap* $\gamma(\mathbf{P}^0_{kk}) \ge \gamma$ *and the stationary measure* $\mu_k := \pi^0_k$ *satisfies* $\rho/M \le \mu_k(x) \le \rho'/M$. *Moreover suppose that the eigenvalue matrix* $\mathbf{V}_k$ *of* $\mathbf{P}^0_{kk}$ *has condition number bounded as* $\kappa(\mathbf{V}_k) = \|\mathbf{V}_k\|_2 \|\mathbf{V}_k^{-1}\|_2 \le \kappa_0\sqrt{M}$, *and the group inverse* $\mathbf{A}^\sharp_k$ *of* $\mathbf{I} - \mathbf{P}^0_{kk}$ *satisfies* $\|\mathbf{A}^\sharp_k\|_\infty \le g_0$ *for constants* $\kappa_0, g_0$. *Then for all* $\varepsilon = o(M^{-1})$,

$$\gamma^\dagger(\mathbf{S}^\varepsilon_{kk}) \ge \frac{\gamma}{2} \quad and \quad \frac{\rho}{2M} \le \pi^\varepsilon_k(x) \le \frac{2\rho'}{M} \quad \forall x \in C_k.$$

*Proof.* By the proportionality of $p^\varepsilon$ in Assumption 2 and (5) we have

$$\begin{aligned}
\|\mathbf{S}^\varepsilon_{kk} - \mathbf{P}^0_{kk}\|_{1,\infty} &\le \|\mathbf{S}^\varepsilon_{kk} - \mathbf{P}^\varepsilon_{kk}\|_{1,\infty} + \|\mathbf{P}^\varepsilon_{kk} - \mathbf{P}^0_{kk}\|_{1,\infty} \\
&\le d_{\text{out}}\varepsilon + \max_{x \in C_k} \sum_{y \in C_k} |p^0(y|x) - p^\varepsilon(y|x)| \\
&\le 2d_{\text{out}}\varepsilon,
\end{aligned}$$

so that $\|\mathbf{S}^\varepsilon_{kk} - \mathbf{P}^0_{kk}\|_2 \le 2\sqrt{M}d_{\text{out}}\varepsilon$ by Lemma B.5. Then by the Bauer-Fike theorem it holds that

$$|\lambda_2(\mathbf{S}^\varepsilon_{kk}) - \lambda_2(\mathbf{P}^0_{kk})| \le \kappa(\mathbf{V}_k)\|\mathbf{S}^\varepsilon_{kk} - \mathbf{P}^0_{kk}\|_2 \le 2\kappa_0 M d_{\text{out}}\varepsilon = o(1),$$

therefore $\gamma^\dagger(\mathbf{S}^\varepsilon_{kk}) \ge \gamma(\mathbf{S}^\varepsilon_{kk}) \ge \frac{\gamma}{2}$ for sufficiently large $M$. Furthermore by the condition number bound in [Meyer](1980) the perturbed stationary distribution satisfies

$$\|\pi^\varepsilon_k - \mu_k\|_\infty \le \|\mathbf{A}^\sharp_k\|_\infty \|\mathbf{S}^\varepsilon_{kk} - \mathbf{P}^0_{kk}\|_\infty \le 2g_0 d_{\text{out}}\varepsilon = o(M^{-1}),$$

proving the second assertion. $\square$

With these results in mind, we can prove the following concentration bound for the reduced chain $\tilde{X}^{k,\varepsilon}$.

**Lemma B.9.** *For all* $x, y \in C_k$ *and* $\delta > 0$ *it holds that* $\mathbb{P}_x(\tilde{\tau}^{k,\varepsilon}_y \ge m) \le \delta$ *as long as*

$$m \ge \frac{8M}{\rho\gamma} \log \frac{1}{\delta} \cdot \log \frac{M}{\rho}. \tag{6}$$

*Proof.* By Proposition B.4, the mixing time of $\tilde{X}^{k,\varepsilon}_t$ is bounded above as

$$t_{\text{mix}} := t_{\text{mix}}\left(\frac{\rho}{2M}\right) \le \frac{1}{\gamma^\dagger(\mathbf{S}^\varepsilon_{kk})}\left(1 + 2\log\frac{M}{\rho} + \log\frac{1}{\min \pi^\varepsilon_k}\right) \le \frac{4}{\gamma}\log\frac{M}{\rho}$$

so that for any $x, y \in C_k$,

$$(s^\varepsilon_{kk})^{t_{\text{mix}}}(y|x) \ge \pi^\varepsilon_k(y) - \left(\pi^\varepsilon_k(y) - (s^\varepsilon_{kk})^{t_{\text{mix}}}(y|x)\right) \ge \frac{\rho}{M} - \|(s^\varepsilon_{kk})^{t_{\text{mix}}} - \pi^\varepsilon_k\|_{\text{TV}} \ge \frac{\rho}{2M}.$$

This implies each step of the $t_{\text{mix}}$-skipped chain $(\tilde{X}_{t_{\text{mix}}t}^{k,\varepsilon})_{t\geq 0}$ is well-mixed, and hence

$$
\begin{aligned}
\mathbb{P}_x(\tilde{\tau}_y^{k,\varepsilon} \geq m) &\leq \sup \prod_{t=1}^{\lfloor m/t_{\text{mix}} \rfloor} \left( 1 - \mathbb{P}_{\tilde{X}_{t_{\text{mix}}(t-1)}^{k,\varepsilon}}(\tilde{X}_{t_{\text{mix}}t}^{k,\varepsilon} = y) \right) \\
&\leq \left( 1 - \frac{\rho}{2M} \right)^{\lfloor m/t_{\text{mix}} \rfloor} \\
&\leq \exp\left( -\frac{\rho m}{2M t_{\text{mix}}} \right) \leq \delta,
\end{aligned}
$$

as was to be shown. $\qquad\square$

### B.3. Detailed Balance of Escape Probabilities

The following 'detailed balance equation' for hitting times, proved in Proposition 3.1 of Betz & Le Roux (2016) for nonreversible Markov chains, will be useful. We reproduce the proof here for convenience.

**Proposition B.10.** *For an irreducible, positive recurrent Markov chain $X$ on a state space $S$ with unique stationary distribution $\pi$, for all $x, y \in S$,*
$$
\pi(x)\mathbb{P}_x(\bar{\tau}_y < \bar{\tau}_x) = \pi(y)\mathbb{P}_y(\bar{\tau}_x < \bar{\tau}_y).
$$

*Proof.* For arbitrary $z \in S$, it holds that
$$
\mathbb{E}_z[\bar{\tau}_x] = \mathbb{E}_z[\min\{\bar{\tau}_x, \bar{\tau}_y\}] + \mathbb{E}_z[(\bar{\tau}_x - \bar{\tau}_y)\mathbf{1}_{\{\bar{\tau}_x > \bar{\tau}_y\}}] = \mathbb{E}_z[\min\{\bar{\tau}_x, \bar{\tau}_y\}] + \mathbb{E}_y[\bar{\tau}_x]\mathbb{P}_z(\bar{\tau}_x > \bar{\tau}_y). \tag{7}
$$

Taking $z = y$ in (7) gives $\mathbb{E}_y[\min\{\bar{\tau}_x, \bar{\tau}_y\}] = \mathbb{E}_y[\bar{\tau}_x]\mathbb{P}_y(\bar{\tau}_x < \bar{\tau}_y)$, and substituting this in (7) with $x, y$ swapped and $z = y$ yields

$$
\frac{1}{\pi(y)} = \mathbb{E}_y[\bar{\tau}_y] = \mathbb{E}_y[\min\{\bar{\tau}_x, \bar{\tau}_y\}] + \mathbb{E}_x[\bar{\tau}_y]\mathbb{P}_y(\bar{\tau}_x < \bar{\tau}_y) = (\mathbb{E}_y[\bar{\tau}_x] + \mathbb{E}_x[\bar{\tau}_y])\mathbb{P}_y(\bar{\tau}_x < \bar{\tau}_y)
$$

or

$$
\pi(y)\mathbb{P}_y(\bar{\tau}_x < \bar{\tau}_y) = \frac{1}{\mathbb{E}_y[\bar{\tau}_x] + \mathbb{E}_x[\bar{\tau}_y]} = \pi(x)\mathbb{P}_x(\bar{\tau}_y < \bar{\tau}_x) \tag{8}
$$

by symmetry. $\qquad\square$

## C. Perturbative Analysis of Metastable Dynamics

### C.1. Quantitative Metastable Dynamics

We first show the following useful bound.

**Lemma C.1.** *There exists $\nu > 0$ such that for all $k \in [K]$ and distinct states $x, y \in C_k$, the unperturbed chain $X^0$ satisfies*
$$
\mathbb{P}_x(\bar{\tau}_y^0 < \bar{\tau}_x^0) \geq \frac{\nu}{\log M}.
$$

*Proof.* From (8) it holds that
$$
\mu_k(y)\mathbb{P}_y(\bar{\tau}_x^0 < \bar{\tau}_y^0) = \frac{1}{\mathbb{E}_y[\bar{\tau}_x^0] + \mathbb{E}_x[\bar{\tau}_y^0]}. \tag{9}
$$

By Assumption 1 it holds that $\mu_k(y) = \Theta(1/M)$. Moreover since the skipped chain $(X_{t_{\text{mix}}t}^0)_{t\geq 0}$ is well-mixed,

$$
\mathbb{P}_x(\tau_y \geq m) \leq \sup \prod_{t=1}^{\lfloor m/t_{\text{mix}} \rfloor} \left( 1 - \mathbb{P}_{X_{t_{\text{mix}}(t-1)}^0}(X_{t_{\text{mix}}t}^0 = m) \right) \leq \exp\left( -\frac{\rho m}{2M t_{\text{mix}}} \right).
$$

It follows that

$$
\mathbb{E}_x[\bar{\tau}_y^0] = \sum_{m=0}^{\infty} \mathbb{P}_x(\tau_y \geq m) \leq \left( 1 - \exp\left( -\frac{\rho}{2M t_{\text{mix}}} \right) \right)^{-1} = O(M \log M)
$$

and $\mathbb{E}_x[\bar{\tau}_y^0] = O(M \log M)$ by symmetry. The statement then follows from (9). $\qquad\square$

To study the cluster transition dynamics, we begin with a decomposition of hitting probabilities, which is closely related to the theory of stochastic complementation (Meyer, 1989). Here, we follow the proof in Betz & Le Roux (2016).

**Lemma C.2.** *For $C \subseteq S$, $x \in S$ and $y \in C$ such that $\mathbb{P}_x(\bar{\tau}_C^\varepsilon < \infty) = 1$, it holds that*

$$\mathbb{P}_x(X_{\bar{\tau}_C^\varepsilon}^\varepsilon = y) = p^\varepsilon(x,y) + \sum_{z \in C^c} \frac{\mathbb{P}_x(\bar{\tau}_z^\varepsilon < \bar{\tau}_C^\varepsilon)}{\mathbb{P}_z(\bar{\tau}_C^\varepsilon < \bar{\tau}_z^\varepsilon)} p^\varepsilon(z,y). \tag{10}$$

*Proof.* If $X_{\bar{\tau}_C^\varepsilon}^\varepsilon = y$, either $X^\varepsilon$ has moved directly from $x$ to $y$ or has first moved to some $z = X_{\bar{\tau}_C^\varepsilon - 1}^\varepsilon \notin C$. Conditioning on the number of returns to $z$ before transitioning to $y$ yields

$$\begin{aligned}
\mathbb{P}_x(X_{\bar{\tau}_C^\varepsilon}^\varepsilon = y) &= p^\varepsilon(x,y) + \sum_{z \in C^c} \sum_{n \geq 1} \mathbb{P}_x(\bar{\tau}_{z,n}^\varepsilon < \bar{\tau}_C^\varepsilon, X_{\bar{\tau}_{z,n}^\varepsilon + 1}^\varepsilon = y) \\
&= p^\varepsilon(x,y) + \sum_{z \in C^c} \sum_{n \geq 1} \mathbb{P}_x(\bar{\tau}_z^\varepsilon < \bar{\tau}_C^\varepsilon) \mathbb{P}_z(\bar{\tau}_z^\varepsilon < \bar{\tau}_C^\varepsilon)^{n-1} p^\varepsilon(z,y) \\
&= p^\varepsilon(x,y) + \sum_{z \in C^c} \frac{\mathbb{P}_x(\bar{\tau}_z^\varepsilon < \bar{\tau}_C^\varepsilon)}{1 - \mathbb{P}_z(\bar{\tau}_z^\varepsilon < \bar{\tau}_C^\varepsilon)} p^\varepsilon(z,y),
\end{aligned}$$

concluding (10). Note that $p^\varepsilon(x,y)$ must be added separately even for $z = x$ as the second term only counts returns to $x$ for time $t > 0$. $\qquad \square$

**Definition C.3** (induced path measure). For $m \in \mathbb{N}$, define the *path measure induced by $X^\varepsilon$ on $S^m$* as

$$\tilde{\mathbb{P}}_x^{\varepsilon,m}(x_{1:m}) := \prod_{i=1}^m p^\varepsilon(x_i | x_{i-1}), \quad x_{1:m} \in S^m, \quad x_0 = x.$$

Similarly to the total variation distance bound between product measures, we have the following result.

**Lemma C.4.** $\|\tilde{\mathbb{P}}_x^{\varepsilon,m} - \tilde{\mathbb{P}}_x^{0,m}\|_{\mathrm{TV}} \leq m d_{\mathrm{out}} \varepsilon$.

*Proof.* Recalling that $\|p^\varepsilon(\cdot | x) - p^0(\cdot | x)\|_{\mathrm{TV}} \leq d_{\mathrm{out}} \varepsilon$ for all $x \in S$,

$$\begin{aligned}
&\|\tilde{\mathbb{P}}_x^{\varepsilon,m} - \tilde{\mathbb{P}}_x^{0,m}\|_{\mathrm{TV}} \\
&= \frac{1}{2} \sum_{x_{1:m}} \left| \mathbb{P}_x^{\varepsilon,m}(x_{1:m}) - \mathbb{P}_x^{0,m}(x_{1:m}) \right| \\
&\leq \frac{1}{2} \sum_{x_{1:m}} \sum_{i=1}^m \left| p^\varepsilon(x_i | x_{i-1}) - p^0(x_i | x_{i-1}) \right| \prod_{j>i} p^\varepsilon(x_j | x_{j-1}) \prod_{j'<i} p^0(x_{j'} | x_{j'-1}) \\
&= \frac{1}{2} \sum_{i=1}^m \sum_{x_{1:i}} \left| p^\varepsilon(x_i | x_{i-1}) - p^0(x_i | x_{i-1}) \right| \prod_{j'<i} p^0(x_{j'} | x_{j'-1}) \\
&\leq d_{\mathrm{out}} \varepsilon \cdot \sum_{i=1}^m \sum_{x_{1:i-1}} \prod_{j'<i} p^0(x_{j'} | x_{j'-1}) \\
&= m d_{\mathrm{out}} \varepsilon.
\end{aligned}$$

$\qquad \square$

**Proposition C.5.** *For all $k \in [K]$ and $\varepsilon \leq O(M^{-1}(\log M)^{-4})$, it holds that*

$$\sup_{x,y \in C_k} \sup_{z \in S} \left| \mathbb{P}_x(\tau_y^\varepsilon < \tau_z^\varepsilon) - \mathbb{P}_x(\tau_y^0 < \tau_z^0) \right| \leq \tilde{O}\left( \frac{1}{(\log M)^3} \right)$$

*and*

$$\sup_{x,y \in C_k} \sup_{z \in S} \left| \mathbb{P}_x(\bar{\tau}_y^\varepsilon < \bar{\tau}_z^\varepsilon) - \mathbb{P}_x(\bar{\tau}_y^0 < \bar{\tau}_z^0) \right| \leq \tilde{O}\left( \frac{1}{(\log M)^3} \right).$$

*Proof.* Since $\bar{\tau}_y^\varepsilon < \infty$ for all $\varepsilon \geq 0$ almost surely for $x, y \in C_k$, the above probabilities are well-defined. We prove only the second inequality. Denote the augmented complements $C^{k,z} := C_k^c \cup \{z\}$ and $C^{k,y,z} := C_k^c \cup \{y, z\}$ for brevity. We divide the event $\{\bar{\tau}_y^\varepsilon < \bar{\tau}_z^\varepsilon\}$ according to whether the chain has been contained in $C_k$ or has first hit some $w \in C_k^c, w \neq z$ before reaching $y$, and bound the magnitude of perturbation of each term:

$$\mathbb{P}_x(\bar{\tau}_y^\varepsilon < \bar{\tau}_z^\varepsilon) = \mathbb{P}_x(\bar{\tau}_y^\varepsilon < \bar{\tau}_{C^{k,z}}^\varepsilon) + \sum_{w \in C_k^c \setminus \{z\}} \mathbb{P}_x(X_{\bar{\tau}_{C^{k,y,z}}^\varepsilon}^\varepsilon = w)\mathbb{P}_w(\bar{\tau}_y^\varepsilon < \bar{\tau}_z^\varepsilon). \tag{11}$$

For the first term, we exploit the fast mixing of $X^\varepsilon$ within $C_k$ to show a concentration result for $\bar{\tau}_y^\varepsilon$, then utilize the path measure perturbation bound. Specifically, for $m$ chosen to satisfy (6), the inequality $m \leq \bar{\tau}_y^\varepsilon < \bar{\tau}_{C^{k,z}}^\varepsilon$ implies $\tilde{X}_t^{k,\varepsilon} = X_t^\varepsilon$ for $t < m$, so that

$$\mathbb{P}_x(\bar{\tau}_y^\varepsilon < \bar{\tau}_{C^{k,z}}^\varepsilon) - \mathbb{P}_x(\bar{\tau}_y^\varepsilon < \bar{\tau}_{C^{k,z}}^\varepsilon \wedge m) = \mathbb{P}_x(m \leq \bar{\tau}_y^\varepsilon < \bar{\tau}_{C^{k,z}}^\varepsilon) \leq \mathbb{P}_x(\tilde{\tau}_y^{k,\varepsilon} \geq m) < \delta. \tag{12}$$

Moreover, define $\Gamma_{y,z}$ to be the set of paths $\gamma$ contained in $C_k$ of length equal to $m$ such that $y$ appears, and first appears before any instance of $z$, that is

$$\Gamma_{y,z} := \big\{\gamma \in C_k^m : \inf\{k \in [m] : \gamma_k = y\} < (m+1) \wedge \inf\{k \in [m] : \gamma_k \in C^{k,z}\}.\big\}$$

It follows that

$$\mathbb{P}_x(\bar{\tau}_y^\varepsilon < \bar{\tau}_{C^{k,z}}^\varepsilon \wedge m) = \tilde{\mathbb{P}}_x^{\varepsilon,m}(\Gamma_{y,z})$$

and hence

$$\begin{aligned}&|\mathbb{P}_x(\bar{\tau}_y^\varepsilon < \bar{\tau}_{C^{k,z}}^\varepsilon \wedge m) - \mathbb{P}_x(\bar{\tau}_y^0 < \bar{\tau}_{C^{k,z}}^0 \wedge m)| \\&= |\tilde{\mathbb{P}}_x^{\varepsilon,m}(\Gamma_{y,z}) - \tilde{\mathbb{P}}_x^{0,m}(\Gamma_{y,z})| \leq \|\tilde{\mathbb{P}}_x^{\varepsilon,m} - \tilde{\mathbb{P}}_x^{0,m}\|_{\mathrm{TV}} \leq md_{\mathrm{out}}\varepsilon\end{aligned}$$

by Lemma C.4.

For the second term, by Lemma C.2 we have for all $w \in C_k^c \setminus \{z\}$

$$\mathbb{P}_x(X_{\bar{\tau}_{C^{k,y,z}}^\varepsilon}^\varepsilon = w) = p^\varepsilon(x, w) + \sum_{u \in C_k \setminus \{y,z\}} \frac{\mathbb{P}_x(\bar{\tau}_u^\varepsilon < \bar{\tau}_{C^{k,y,z}}^\varepsilon)}{\mathbb{P}_u(\bar{\tau}_{C^{k,y,z}}^\varepsilon < \bar{\tau}_u^\varepsilon)} p^\varepsilon(u, w).$$

The denominator can be lower bounded via a path measure argument similar to before:

$$\begin{aligned}\mathbb{P}_u(\bar{\tau}_{C^{k,y,z}}^\varepsilon < \bar{\tau}_u^\varepsilon) &\geq \mathbb{P}_u(\bar{\tau}_y^\varepsilon < \bar{\tau}_u^\varepsilon \wedge m) \\&\geq \mathbb{P}_u(\bar{\tau}_y^0 < \bar{\tau}_u^0 \wedge m) - \|\tilde{\mathbb{P}}_x^{\varepsilon,m} - \tilde{\mathbb{P}}_x^{0,m}\|_{\mathrm{TV}} \\&\geq \mathbb{P}_u(\bar{\tau}_y^0 < \bar{\tau}_u^0) - \mathbb{P}_u(\bar{\tau}_y^0 \geq m) - \|\tilde{\mathbb{P}}_x^{\varepsilon,m} - \tilde{\mathbb{P}}_x^{0,m}\|_{\mathrm{TV}} \\&\geq \frac{\nu}{\log M} - \delta - md_{\mathrm{out}}\varepsilon \geq \frac{\nu}{2\log M}\end{aligned} \tag{13}$$

by Lemma C.1, as long as $\delta, m\varepsilon = o((\log M)^{-1})$. It follows that

$$\mathbb{P}_x(X_{\bar{\tau}_{C^{k,y,z}}^\varepsilon}^\varepsilon = w) \leq p^\varepsilon(x, w) + \frac{2\log M}{\nu} \sum_{u \in C_k \setminus \{y,z\}} p^\varepsilon(u, w)$$

and

$$\begin{aligned}&\sum_{w \in C_k^c \setminus \{z\}} \mathbb{P}_x(X_{\bar{\tau}_{C^{k,y,z}}^\varepsilon}^\varepsilon = w)\mathbb{P}_w(\bar{\tau}_y^\varepsilon < \bar{\tau}_z^\varepsilon) \\&\leq \sum_{w \in C_k^c \setminus \{z\}} p^\varepsilon(x, w) + \frac{2\log M}{\nu} \sum_{u \in C_k \setminus \{y,z\}} \sum_{w \in C_k^c \setminus \{z\}} p^\varepsilon(u, w)\end{aligned}$$

$$\leq \left(1 + \frac{2n_{\text{out}} \log M}{\nu}\right) d_{\text{out}}\varepsilon. \tag{14}$$

Now taking $\delta = O(M\varepsilon \log M)$ and $\varepsilon \leq O(M^{-1}(\log M)^{-4})$, we can verify that $\delta = O((\log M)^{-3})$ and

$$m\varepsilon = O\left(M\varepsilon \log \frac{1}{M\varepsilon} \cdot \log M\right) = O\left(\frac{\log \log M}{(\log M)^3}\right).$$

Combining (11), (12) and (14), we conclude:

$$\left|\mathbb{P}_x(\bar{\tau}_y^\varepsilon < \bar{\tau}_z^\varepsilon) - \mathbb{P}_x(\bar{\tau}_y^0 < \bar{\tau}_z^0)\right| \leq md_{\text{out}}\varepsilon + \delta + O(\log M) \cdot d_{\text{out}}\varepsilon = \widetilde{O}\left(\frac{1}{(\log M)^3}\right),$$

as was to be shown. $\qquad\square$

As a corollary, we obtain:

**Corollary C.6** (Proposition 2.1 restated). *Any subset $S_\circ = \{x_1, \cdots, x_K\} \subset S$ of cluster representatives $x_k \in C_k$ constitutes a metastable system for $X^\varepsilon$ in the sense of* (1) *as $M \to \infty$.*

*Proof.* For $y \in C_k \setminus \{x_k\}$, it holds that

$$\mathbb{P}_y(\bar{\tau}_{S_\circ}^\varepsilon < \bar{\tau}_y^\varepsilon) \geq \mathbb{P}_y(\bar{\tau}_{x_k}^\varepsilon < \bar{\tau}_y^\varepsilon) \geq \frac{\nu}{2 \log M}$$

similarly to (13). On the other hand, for $x_k \in S_\circ$ it follows from Proposition C.5 that

$$\mathbb{P}_{x_k}(\bar{\tau}_{S_\circ \setminus \{x_k\}}^\varepsilon < \bar{\tau}_{x_k}^\varepsilon) \leq \mathbb{P}_{x_k}(\bar{\tau}_{S_\circ \setminus \{x_k\}}^0 < \bar{\tau}_{x_k}^0) + \widetilde{O}\left(\frac{1}{(\log M)^3}\right),$$

and hence (1) follows. $\qquad\square$

Now let us study the convergence of the perturbed stationary distributions. Let $\pi^\varepsilon$ for $\varepsilon > 0$ denote the unique stationary distribution of $X^\varepsilon$ on $S$. By the coupling theorem (Meyer, 1989, Theorem 4.1),

$$\pi^\varepsilon = (\xi_1 \pi_1^\varepsilon \ \cdots \ \xi_K \pi_K^\varepsilon) \tag{15}$$

where the coupling factors $\xi_k = \pi^\varepsilon(C_k)$. We then obtain the following corollary of Proposition C.5.

**Corollary C.7.** *For all $k \in [K]$, it holds that*

$$\sup_{x \in C_k} \left|\frac{\pi_k^\varepsilon(x)}{\mu_k(x)} - 1\right| \leq \widetilde{O}\left(\frac{1}{\log M}\right).$$

We remark that compared to the straightforward perturbation bound in Proposition B.8, this approach does not require reversibility nor an explicit condition number bound.

*Proof.* For all $x, y \in C_k$, by Proposition B.10 applied to $X^\varepsilon$ on $S$ and $X^0$ on $C_k$,

$$\frac{\pi_k^\varepsilon(x)}{\pi_k^\varepsilon(y)} = \frac{\pi^\varepsilon(x)}{\pi^\varepsilon(y)} = \frac{\mathbb{P}_y(\bar{\tau}_x^\varepsilon < \bar{\tau}_y^\varepsilon)}{\mathbb{P}_x(\bar{\tau}_y^\varepsilon < \bar{\tau}_x^\varepsilon)}, \quad \frac{\mu_k(x)}{\mu_k(y)} = \frac{\mathbb{P}_y(\bar{\tau}_x^0 < \bar{\tau}_y^0)}{\mathbb{P}_x(\bar{\tau}_y^0 < \bar{\tau}_x^0)}.$$

Recall that $\mathbb{P}_x(\bar{\tau}_y^0 < \bar{\tau}_x^0) \geq \frac{\nu}{\log M}$ by Lemma C.1 and moreover $\mathbb{P}_x(\bar{\tau}_y^\varepsilon < \bar{\tau}_x^\varepsilon) \geq \frac{\nu}{2 \log M}$ by repeating the argument in (13). Therefore,

$$\left|\frac{\pi_k^\varepsilon(x)}{\pi_k^\varepsilon(y)} - \frac{\mu_k(x)}{\mu_k(y)}\right| = \left|\frac{\mathbb{P}_y(\bar{\tau}_x^\varepsilon < \bar{\tau}_y^\varepsilon)}{\mathbb{P}_x(\bar{\tau}_y^\varepsilon < \bar{\tau}_x^\varepsilon)} - \frac{\mathbb{P}_y(\bar{\tau}_x^0 < \bar{\tau}_y^0)}{\mathbb{P}_x(\bar{\tau}_y^0 < \bar{\tau}_x^0)}\right|$$

$$\leq \frac{\left|\mathbb{P}_x(\bar{\tau}_y^\varepsilon < \bar{\tau}_x^\varepsilon) - \mathbb{P}_x(\bar{\tau}_y^0 < \bar{\tau}_x^0)\right| + \left|\mathbb{P}_y(\bar{\tau}_x^\varepsilon < \bar{\tau}_y^\varepsilon) - \mathbb{P}_y(\bar{\tau}_x^0 < \bar{\tau}_y^0)\right|}{\mathbb{P}_x(\bar{\tau}_y^\varepsilon < \bar{\tau}_x^\varepsilon)\mathbb{P}_x(\bar{\tau}_y^0 < \bar{\tau}_x^0)}$$

$$\leq \frac{4(\log M)^2}{\nu^2} \cdot \widetilde{O}\left(\frac{1}{(\log M)^3}\right).$$

By Assumption 1 we have that $\mu_k(y)/\mu_k(x)$ is bounded for all $x, y \in C_k$ and hence

$$\left|\frac{\pi_k^\varepsilon(x)}{\mu_k(x)} - 1\right| \leq \sum_{y \in C_k} \left|\frac{\pi_k^\varepsilon(x)}{\mu_k(x)}\mu_k(y) - \pi_k^\varepsilon(y)\right|$$

$$= \sum_{y \in C_k} \frac{\mu_k(y)}{\mu_k(x)}\pi_k^\varepsilon(y)\left|\frac{\pi_k^\varepsilon(x)}{\pi_k^\varepsilon(y)} - \frac{\mu_k(x)}{\mu_k(y)}\right| \leq \widetilde{O}\left(\frac{1}{\log M}\right).$$

$\square$

## C.2. Perturbative Analysis of Metastable Chain

We proceed to study the behavior of the meta-chain $X_\star^\varepsilon$ with transition probabilities $q_\star^\varepsilon$ defined in (2). It can be shown that $X_\star^\varepsilon$ is asymptotically reversible with respect to the measure induced by $\pi^\varepsilon$:

**Proposition C.8.** *For all $k, \ell \in [K]$ with $k \neq \ell$ it holds that*

$$\frac{\pi^\varepsilon(C_k)q_\star^\varepsilon(C_\ell|C_k)}{\pi^\varepsilon(C_\ell)q_\star^\varepsilon(C_k|C_\ell)} = 1 + \widetilde{O}\left(\frac{1}{\log M}\right).$$

*Proof.* First note that for $x \in C_k$, $y \in C_\ell$ with $k \neq \ell$, by Proposition C.5,

$$0 \leq 1 - \frac{\mathbb{P}_x(\bar{\tau}_y^\varepsilon < \bar{\tau}_x^\varepsilon)}{\mathbb{P}_x(\bar{\tau}_{C_\ell}^\varepsilon < \bar{\tau}_x^\varepsilon)}$$

$$= \frac{1}{\mathbb{P}_x(\bar{\tau}_{C_\ell}^\varepsilon < \bar{\tau}_x^\varepsilon)} \sum_{z \in C_\ell} \mathbb{P}_x(\bar{\tau}_{C_\ell}^\varepsilon < \bar{\tau}_x^\varepsilon, X_{\bar{\tau}_{C_\ell}^\varepsilon}^\varepsilon = z)\mathbb{P}_z(\tau_x^\varepsilon < \tau_y^\varepsilon)$$

$$\leq \frac{1}{\mathbb{P}_x(\bar{\tau}_{C_\ell}^\varepsilon < \bar{\tau}_x^\varepsilon)} \sum_{z \in C_\ell} \mathbb{P}_x(\bar{\tau}_{C_\ell}^\varepsilon < \bar{\tau}_x^\varepsilon, X_{\bar{\tau}_{C_\ell}^\varepsilon}^\varepsilon = z) \cdot \sup_{z \in C_\ell} \left|\mathbb{P}_z(\tau_x^\varepsilon < \tau_y^\varepsilon) - \mathbb{P}_z(\tau_x^0 < \tau_y^0)\right|$$

$$\leq \widetilde{O}\left(\frac{1}{(\log M)^3}\right).$$

By the definition of $q_\star^\varepsilon$, the coupling equation (15) and Corollary C.7, it follows that

$$\pi^\varepsilon(C_k)q_\star^\varepsilon(C_\ell|C_k) = \pi^\varepsilon(C_k) \sum_{x \in C_k} \mu_k(x)^2\mathbb{P}_x(\bar{\tau}_{C_\ell}^\varepsilon < \bar{\tau}_x^\varepsilon)$$

$$= \pi^\varepsilon(C_k) \sum_{x \in C_k} \pi_k^\varepsilon(x)\mu_k(x)\mathbb{P}_x(\bar{\tau}_{C_\ell}^\varepsilon < \bar{\tau}_x^\varepsilon)$$

$$+ \pi^\varepsilon(C_k) \sum_{x \in C_k} \mu_k(x)^2\left(1 - \frac{\pi_k^\varepsilon(x)}{\mu_k(x)}\right)\mathbb{P}_x(\bar{\tau}_{C_\ell}^\varepsilon < \bar{\tau}_x^\varepsilon)$$

$$= \sum_{x \in C_k} \pi^\varepsilon(x)\mu_k(x)\mathbb{P}_x(\bar{\tau}_{C_\ell}^\varepsilon < \bar{\tau}_x^\varepsilon) + \pi^\varepsilon(C_k)q_\star^\varepsilon(C_\ell|C_k) \cdot \widetilde{O}\left(\frac{1}{\log M}\right).$$

We expand the first term further as

$$\sum_{x \in C_k} \pi^\varepsilon(x)\mu_k(x)\mathbb{P}_x(\bar{\tau}_{C_\ell}^\varepsilon < \bar{\tau}_x^\varepsilon)$$

$$= \sum_{x \in C_k} \sum_{y \in C_\ell} \pi^\varepsilon(x)\mu_k(x)\mu_\ell(y)\mathbb{P}_x(\bar{\tau}_y^\varepsilon < \bar{\tau}_x^\varepsilon)$$

$$+ \sum_{x \in C_k} \sum_{y \in C_\ell} \pi^\varepsilon(x) \mu_k(x) \mu_\ell(y) \mathbb{P}_x(\bar{\tau}^\varepsilon_{C_\ell} < \bar{\tau}^\varepsilon_x) \left( 1 - \frac{\mathbb{P}_x(\bar{\tau}^\varepsilon_y < \bar{\tau}^\varepsilon_x)}{\mathbb{P}_x(\bar{\tau}^\varepsilon_{C_\ell} < \bar{\tau}^\varepsilon_x)} \right)$$

$$= \sum_{x \in C_k} \sum_{y \in C_\ell} \pi^\varepsilon(x) \mu_k(x) \mu_\ell(y) \mathbb{P}_x(\bar{\tau}^\varepsilon_y < \bar{\tau}^\varepsilon_x)$$

$$+ \sum_{x \in C_k} \pi^\varepsilon(x) \mu_k(x) \mathbb{P}_x(\bar{\tau}^\varepsilon_{C_\ell} < \bar{\tau}^\varepsilon_x) \cdot \widetilde{O} \left( \frac{1}{(\log M)^3} \right).$$

Together, we have shown that

$$\pi^\varepsilon(C_k) q^\varepsilon_\star(C_\ell | C_k) = \sum_{x \in C_k} \sum_{y \in C_\ell} \pi^\varepsilon(x) \mu_k(x) \mu_\ell(y) \mathbb{P}_x(\bar{\tau}^\varepsilon_y < \bar{\tau}^\varepsilon_x) \cdot \left( 1 + \widetilde{O} \left( \frac{1}{\log M} \right) \right),$$

and by symmetry

$$\pi^\varepsilon(C_\ell) q^\varepsilon_\star(C_k | C_\ell) = \sum_{x \in C_k} \sum_{y \in C_\ell} \pi^\varepsilon(y) \mu_k(x) \mu_\ell(y) \mathbb{P}_y(\bar{\tau}^\varepsilon_x < \bar{\tau}^\varepsilon_y) \cdot \left( 1 + \widetilde{O} \left( \frac{1}{\log M} \right) \right).$$

Finally, since

$$\sum_{x \in C_k} \sum_{y \in C_\ell} \mu_k(x) \mu_\ell(y) \cdot \pi^\varepsilon(x) \mathbb{P}_x(\bar{\tau}^\varepsilon_y < \bar{\tau}^\varepsilon_x) = \sum_{x \in C_k} \sum_{y \in C_\ell} \mu_k(x) \mu_\ell(y) \cdot \pi^\varepsilon(y) \mathbb{P}_y(\bar{\tau}^\varepsilon_x < \bar{\tau}^\varepsilon_y)$$

due to Proposition B.10, we conclude the desired statement. $\qquad \square$

Together with Assumptions 1, 4 and (15), this immediately implies:

**Corollary C.9.** *For all $k \in [K]$ and $x \in S$ it holds that $\pi^\varepsilon(C_k) = \Theta(1/K)$ and $\pi^\varepsilon(x) = \Theta(1/KM)$.*

Moreover, $q^\varepsilon_\star(\cdot | C_k)$ serves as an approximation of the escape probabilities from any $x \in C_k$, weighted by the stationary measure.

**Proposition C.10.** *For $k, \ell \in [K]$ with $k \neq \ell$ it holds that*

$$\sup_{x \in C_k} \left| \frac{\mu_k(x) \mathbb{P}_x(\bar{\tau}^\varepsilon_{C_\ell} < \bar{\tau}^\varepsilon_x)}{q^\varepsilon_\star(C_\ell | C_k)} - 1 \right| = \widetilde{O} \left( \frac{1}{\log M} \right) \tag{16}$$

*and*

$$\sup_{x \in C_k, y \in C_\ell} \left| \frac{\mu_k(x) \mathbb{P}_x(\bar{\tau}^\varepsilon_y < \bar{\tau}^\varepsilon_x)}{q^\varepsilon_\star(C_\ell | C_k)} - 1 \right| = \widetilde{O} \left( \frac{1}{\log M} \right). \tag{17}$$

*Proof.* Similarly to the proof of Proposition C.8, for any $y \in C_\ell$ we can successively transform via Proposition B.10:

$$\mu_k(x) \mathbb{P}_x(\bar{\tau}^\varepsilon_{C_\ell} < \bar{\tau}^\varepsilon_x) = \mu_k(x) \mathbb{P}_x(\bar{\tau}^\varepsilon_y < \bar{\tau}^\varepsilon_x) \cdot \left( 1 + \widetilde{O} \left( \frac{1}{\log M} \right) \right)$$

$$= \frac{\pi^\varepsilon(x)}{\pi^\varepsilon(C_k)} \mathbb{P}_x(\bar{\tau}^\varepsilon_y < \bar{\tau}^\varepsilon_x) \cdot \left( 1 + \widetilde{O} \left( \frac{1}{\log M} \right) \right)$$

$$= \frac{\pi^\varepsilon(y)}{\pi^\varepsilon(C_k)} \mathbb{P}_y(\bar{\tau}^\varepsilon_x < \bar{\tau}^\varepsilon_y) \cdot \left( 1 + \widetilde{O} \left( \frac{1}{\log M} \right) \right)$$

$$= \frac{\pi^\varepsilon(C_\ell)}{\pi^\varepsilon(C_k)} \pi^\varepsilon_k(y) \mathbb{P}_y(\bar{\tau}^\varepsilon_{C_k} < \bar{\tau}^\varepsilon_y) \cdot \left( 1 + \widetilde{O} \left( \frac{1}{\log M} \right) \right).$$

Since the last term is independent of $x$, we also have

$$q^\varepsilon_\star(C_\ell | C_k) = \sum_{x \in C_k} \mu_k(x)^2 \mathbb{P}_x(\bar{\tau}^\varepsilon_{C_\ell} < \bar{\tau}^\varepsilon_x)$$

$$= \frac{\pi^\varepsilon(C_\ell)}{\pi^\varepsilon(C_k)} \pi_k^\varepsilon(y) \mathbb{P}_y(\bar{\tau}_{C_k}^\varepsilon < \bar{\tau}_y^\varepsilon) \cdot \left(1 + \widetilde{O}\left(\frac{1}{\log M}\right)\right)$$

$$= \mu_k(x) \mathbb{P}_x(\bar{\tau}_{C_\ell}^\varepsilon < \bar{\tau}_x^\varepsilon) \cdot \left(1 + \widetilde{O}\left(\frac{1}{\log M}\right)\right)$$

for any $x \in C_k$, verifying (16). The proof for (17) is identical. $\qquad\square$

As a corollary, we obtain the promised justification of Assumption 4.

**Corollary C.11.** *If there exists a sparse edge from $C_k$ to $C_\ell$, it holds that $q_\star^\varepsilon(C_\ell | C_k) = \widetilde{\Omega}(\varepsilon/M)$.*

*Proof.* Fix $x \in C_k$ and let $(y, z) \in E^s$ with $y \in C_k, z \in C_\ell$. The event $\{\bar{\tau}_{C_\ell}^\varepsilon < \bar{\tau}_x^\varepsilon\}$ occurs if $y$ is hit before returning to $x$ and the edge to $z$ is immediately taken, so that

$$\mathbb{P}_x(\bar{\tau}_{C_\ell}^\varepsilon < \bar{\tau}_x^\varepsilon) \geq p^\varepsilon(z|y) \mathbb{P}_x(\bar{\tau}_y^\varepsilon < \bar{\tau}_x^\varepsilon) \geq \frac{\nu c \varepsilon}{2 \log M}.$$

Hence by Proposition C.10 we obtain

$$q_\star^\varepsilon(C_\ell | C_k) = \mu_k(x) \mathbb{P}_x(\bar{\tau}_{C_\ell}^\varepsilon < \bar{\tau}_x^\varepsilon) \left(1 + \widetilde{O}\left(\frac{1}{\log M}\right)\right) \geq \Omega\left(\frac{\varepsilon}{M \log M}\right).$$

$\qquad\square$

### C.3. Hitting Time Analysis

To prove Theorem 3.2, we first derive the expected escape time of a single cluster.

**Lemma C.12.** *For all $k \in [K]$ and $x \in C_k$, it holds that*

$$\mathbb{E}_x[\tau_{C_k^c}^\varepsilon] = \widetilde{\Theta}\left(\frac{M}{\varepsilon}\right).$$

*Proof.* Recall from Lemma B.9 that the mixing time of $\tilde{X}^{k,\varepsilon}$ is $t_{\mathrm{mix}} = O(\log M)$. Also denote the set of states in $C_k$ with outbound edges as $D_k := \{x \in C_k : \exists y \notin C_k, (x, y) \in E_s\}$, so that $1 \leq |D_k| \leq n_{\mathrm{out}}$ By Assumption 2. Since $\tau_{C_k^c}^\varepsilon > m$ implies that $\tilde{X}_t^{k,\varepsilon} = X_t^\varepsilon$ for $t \leq m$ and that a sparse edge was not taken at each state of the skipped subchain $\tilde{X}_{t_{\mathrm{mix}}t}^{k,\varepsilon}$ up to $t = \lfloor m/t_{\mathrm{mix}} \rfloor$, it follows that

$$\mathbb{P}_x(\tau_{C_k^c}^\varepsilon > m) \leq \sup \prod_{t=1}^{\lfloor m/t_{\mathrm{mix}} \rfloor} \mathbb{P}_{X_{t_{\mathrm{mix}}(t-1)+1}^\varepsilon}(X_{t_{\mathrm{mix}}t+1}^\varepsilon \in C_k)$$

$$\leq \sup \prod_{t=1}^{\lfloor m/t_{\mathrm{mix}} \rfloor} \left(1 - \mathbb{P}_{X_{t_{\mathrm{mix}}(t-1)+1}^\varepsilon}(\tilde{X}_{t_{\mathrm{mix}}t}^{k,\varepsilon} \in D_k) \cdot \mathbb{P}_{\tilde{X}_{t_{\mathrm{mix}}t}^{k,\varepsilon}}(X_{t_{\mathrm{mix}}t+1}^{k,\varepsilon} \notin C_k)\right)$$

$$\leq \prod_{t=1}^{\lfloor m/t_{\mathrm{mix}} \rfloor} \left(1 - \frac{\rho |D_k|}{2M} \cdot c\varepsilon\right)$$

$$\leq \exp\left(-\frac{\rho c \varepsilon m}{2M t_{\mathrm{mix}}}\right).$$

This yields the upper bound

$$\mathbb{E}_x[\tau_{C_k^c}^\varepsilon] = \sum_{m=0}^\infty \mathbb{P}_x(\tau_{C_k^c}^\varepsilon > m) \leq \left(1 - \exp\left(-\frac{\rho c \varepsilon}{2M t_{\mathrm{mix}}}\right)\right)^{-1} \leq O\left(\frac{M \log M}{\varepsilon}\right).$$

For the lower bound, consider the partition of $(\tilde{X}_t^{k,\varepsilon})_{t \geq 0}$ into the union of skipped and shifted subchains $(\tilde{X}_{t_{\mathrm{mix}}t+u}^{k,\varepsilon})_{t \geq 0}$ for $0 \leq u < t_{\mathrm{mix}}$. Suppose that $m \geq 2 t_{\mathrm{mix}}$, so each subchain has length at least 2, and all transition probabilities of each subchain is $\Theta(1/M)$ by Assumption 1. Since not taking a sparse edge at each step of all subchains implies $\tau_{C_k^c}^\varepsilon > m$,

$$\mathbb{P}_x(\tau_{C_k^c}^\varepsilon > m)$$

$$\geq \inf \prod_{u=0}^{t_{\mathrm{mix}}-1} \prod_{t=1}^{\lfloor m/t_{\mathrm{mix}}\rfloor} \left( 1 - \mathbb{P}_{\tilde{X}^{k,\varepsilon}_{t_{\mathrm{mix}}(t-1)+u+1}} (\tilde{X}^{k,\varepsilon}_{t_{\mathrm{mix}}t+u} \in D_k) \cdot \mathbb{P}_{\tilde{X}^{k,\varepsilon}_{t_{\mathrm{mix}}t+u}} (X^{\varepsilon}_{t_{\mathrm{mix}}t+u+1} \notin C_k) \right)$$

$$\geq \prod_{u=0}^{t_{\mathrm{mix}}-1} \prod_{t=1}^{\lfloor m/t_{\mathrm{mix}}\rfloor} \left( 1 - \Theta\left( \frac{|D_k|}{M} \cdot d_{\mathrm{out}}\varepsilon \right) \right)$$

$$\geq \left( 1 - \Theta\left( \frac{\varepsilon}{M} \right) \right)^{m-t_{\mathrm{mix}}}.$$

Note that while the dependency on $t_{\mathrm{mix}}$ does not explicitly appear in the bound, $t_{\mathrm{mix}}$ still needs to be small enough to argue that the states of each subchain for $t \geq 1$ exist and are sufficiently mixed. Hence it follows that

$$\mathbb{E}_x[\tau^{\varepsilon}_{C_k^c}] = \sum_{m=0}^{\infty} \mathbb{P}_x(\tau^{\varepsilon}_{C_k^c} > m) \geq \Omega\left( \frac{M}{\varepsilon} \right) - 2t_{\mathrm{mix}} \geq \Omega\left( \frac{M}{\varepsilon} \right),$$

which concludes the statement. $\qquad\square$

**Theorem C.13** (Theorem 3.2 restated). *Under Assumptions 1-4, it holds for all $\varepsilon \leq \varepsilon_{\max} := \Theta(M^{-1}(\log M)^{-4})$ that*

$$\mathbb{E}_{(X_{\mathrm{in}}, X_{\mathrm{out}}) \sim \mathcal{D}}[\mathbb{E}_{X_{\mathrm{in}}}[\tau^{\varepsilon}_{X_{\mathrm{out}}}]] = \widetilde{\Theta}\left( \frac{KM}{\varepsilon} \right).$$

*Proof.* Suppose $X_{\mathrm{in}} \in C_k, X_{\mathrm{out}} \in C_\ell$ with $k \neq \ell$. For the upper bound, by (8) it holds that

$$\mathbb{E}_{X_{\mathrm{in}}}[\tau^{\varepsilon}_{X_{\mathrm{out}}}] = \mathbb{E}_{X_{\mathrm{in}}}[\bar{\tau}^{\varepsilon}_{X_{\mathrm{out}}}] \leq \mathbb{E}_{X_{\mathrm{in}}}[\bar{\tau}^{\varepsilon}_{X_{\mathrm{out}}}] + \mathbb{E}_{X_{\mathrm{out}}}[\bar{\tau}^{\varepsilon}_{X_{\mathrm{in}}}] = \frac{1}{\pi^{\varepsilon}(X_{\mathrm{in}}) \mathbb{P}_{X_{\mathrm{in}}}(\bar{\tau}^{\varepsilon}_{X_{\mathrm{out}}} < \bar{\tau}^{\varepsilon}_{X_{\mathrm{in}}})}.$$

Combining (15), Corollary C.7 and Proposition C.10 yields

$$\pi^{\varepsilon}(X_{\mathrm{in}}) \mathbb{P}_{X_{\mathrm{in}}}(\bar{\tau}^{\varepsilon}_{X_{\mathrm{out}}} < \bar{\tau}^{\varepsilon}_{X_{\mathrm{in}}}) = \pi^{\varepsilon}(C_k) \cdot \frac{\pi^{\varepsilon}_k(X_{\mathrm{in}})}{\mu_k(X_{\mathrm{in}})} \cdot \mu_k(X_{\mathrm{in}}) \mathbb{P}_{X_{\mathrm{in}}}(\bar{\tau}^{\varepsilon}_{X_{\mathrm{out}}} < \bar{\tau}^{\varepsilon}_{X_{\mathrm{in}}})$$

$$= \pi^{\varepsilon}(C_k) q^{\varepsilon}_{\star}(C_\ell|C_k) \cdot \left( 1 + \widetilde{O}\left( \frac{1}{\log M} \right) \right)$$

$$= O\left( \frac{\varepsilon}{KM} \right),$$

where the last line follows from Corollary C.9 and Assumption 4.

For the lower bound, define the sequence of increasing stopping times $(\sigma_n)_{n \geq 0}$ as

$$\sigma_0 := 0, \quad \sigma_n := \min\{t > \sigma_{n-1} : (X^{\varepsilon}_{t-1}, X^{\varepsilon}_t) \in E_s\}.$$

Then defining the minimum number of cluster transitions to reach $X_{\mathrm{out}}$ as

$$N = N(X_{\mathrm{in}}, X_{\mathrm{out}}) := \min\{|X_{0:T} \cap E_s| : X_0 = X_{\mathrm{in}}, X_T = X_{\mathrm{out}}, (X_{t-1}, X_t) \in E \ \forall t\},$$

it holds that $\tau^{\varepsilon}_{X_{\mathrm{out}}} \geq \sigma_N$. Moreover denoting the cluster containing $X^{\varepsilon}_{\sigma_{t-1}}$ as $C[t]$, by Lemma C.12 we have

$$\mathbb{E}_{X_{\mathrm{in}}}[\sigma_N] = \sum_{t=1}^{N} \mathbb{E}_{X^{\varepsilon}_{\sigma_{t-1}}}[\tau^{\varepsilon}_{C[t]}] \geq \widetilde{\Theta}\left( \frac{MN}{\varepsilon} \right),$$

and hence

$$\mathbb{E}_{(X_{\mathrm{in}}, X_{\mathrm{out}}) \sim \mathcal{D}}[\mathbb{E}_{X_{\mathrm{in}}}[\tau^{\varepsilon}_{X_{\mathrm{out}}}]] \geq \widetilde{\Theta}\left( \frac{KM}{\varepsilon} \right)$$

since $\mathbb{E}[N] = \Omega(K)$ by Assumption 3. $\qquad\square$

# D. Proofs for Optimization Dynamics

## D.1. Analysis of Pretraining Dynamics

**Theorem D.1.** *Let $X_0 \sim \mathrm{Unif}(S)$ or $X_0 \sim \pi^\varepsilon$ and $X_1 \sim p^\varepsilon(\cdot|X_0)$ be random samples from the Markov chain $X^\varepsilon$. Then:*

(1) *The sequence of gradient descent iterates $(\mathbf{W}^{(t)})_{t\geq 0}$ for cross-entropy loss*

$$L_{\mathrm{pre}}(\mathbf{W}) = \mathbb{E}_{X_0,X_1}[-\log \hat{p}_{\mathbf{W}}(X_1|X_0)]$$

*with initialization $\mathbf{W}^{(0)} = \mathbf{0}$ and suitable learning rate converges with respect to the learned transition probabilities as*

$$\sup_{1\leq i,j\leq S} |\hat{p}_{ij}^{(T)} - p_{ij}^\varepsilon| = O\left(\sqrt{\frac{KM^2}{T}}\log\frac{KT}{M\varepsilon}\right). \tag{18}$$

(2) *After $T_1 = \widetilde{O}(KM^2\varepsilon^{-2})$ steps, by setting $w_{ij} \leftarrow -\infty$ if $\hat{p}_{ij}^{(T_1)}$ is below a threshold $c_{\mathrm{thres}}\varepsilon$ it holds for the resulting model $\hat{p}$ that $\hat{p}_{ij} = 0$ iff $p_{ij}^\varepsilon = 0$ and*

$$p_{ij}^\varepsilon - o(\varepsilon) \leq \hat{p}_{ij} \leq p_{ij}^\varepsilon + o(1) \tag{19}$$

*holds uniformly for all $j$ such that $p_{ij}^\varepsilon \neq 0$.*

(3) *After thresholding, the learned transition probabilities converge linearly as*

$$\sup_{1\leq i,j\leq S} |\hat{p}_{ij}^{(T_1+T)} - p_{ij}^\varepsilon| = \exp(-\Omega(\varepsilon^2 T)) \cdot O(\log\varepsilon^{-1}). \tag{20}$$

*Proof.* For part (1), we utilize the proof technique of Ji & Telgarsky (2019), Theorem 3.1 for logistic regression. Suppose $X_0 \sim \mu$ where $\mu$ is any distribution such that $\mu_i = \Theta(1/KM)$ for all states $i$. If $\mu = \pi^\varepsilon$, we will show that $\pi^\varepsilon(x) = \Theta(1/KM)$ for all $x \in S$ in Corollary C.9. The categorical cross-entropy loss can be written as

$$L_{\mathrm{pre}}(\mathbf{W}) = \mathbb{E}_{X_0,X_1}[-\log\hat{p}_{\mathbf{W}}(X_1|X_0)] = \sum_i \mu_i L_i(\mathbf{W}_{i*})$$

where

$$L_i(\mathbf{W}_{i*}) = -\sum_j p_{ij}^\varepsilon w_{ij} + \log\sum_j \exp w_{ij}.$$

Note that each $L_i$ is convex and

$$\inf L_i = -\sum_j p_{ij}^\varepsilon \log p_{ij}^\varepsilon = H(p^\varepsilon(\cdot|e_i))$$

is the entropy of $X_1$ given $X_0 = e_i$. The gradient of $L_i$ is given as $(\nabla L_i)_j = \hat{p}_{\mathbf{W}}(e_j|e_i) - p_{ij}^\varepsilon$. Since the softmax operator is 1-Lipschitz, it follows that $\nabla L_i$ is also 1-Lipschitz,

$$\|\nabla L_i(\mathbf{W}_{i*}) - \nabla L_i(\mathbf{W}_{i*}')\|^2 = \sum_j \left(\frac{\exp w_{ij}}{\sum_k \exp w_{ik}} - \frac{\exp w_{ij}'}{\sum_k \exp w_{ik}'}\right)^2 \leq \|\mathbf{W}_{i*} - \mathbf{W}_{i*}'\|^2.$$

Choose the learning rate $\eta = \Theta(KM)$ such that $\eta_0 \leq \mu_i\eta \leq 1$ for some $\eta_0 > 0$. Then rewriting gradient descent of $\mathbf{W}_{i*}$ as gradient descent with respect to $L_i$,

$$\mathbf{W}_{i*}^{(t+1)} = \mathbf{W}_{i*}^{(t)} - \eta\nabla_{\mathbf{W}_{i*}}L(\mathbf{W}_{i*}^{(t)}) = \mathbf{W}_{i*}^{(t)} - \mu_i\eta\nabla L_i(\mathbf{W}_{i*}^{(t)})$$

we have the well-known guarantee

$$L_i(\mathbf{W}_{i*}^{(t+1)}) \leq L_i(\mathbf{W}_{i*}^{(t)}) - \frac{\eta_0}{2}\|\nabla L_i(\mathbf{W}_{i*}^{(t)})\|^2. \tag{21}$$

Define the reference matrix $\mathbf{Z} \in \mathbb{R}^{|S|\times|S|}$ componentwise as

$$z_{ij} := \log\left(\frac{(1-\delta)|S|}{\delta}p_{ij}^\varepsilon + 1\right),$$

for $\delta > 0$ to be determined. It holds that

$$\|\mathbf{Z}_{i*}\|^2 = \sum_{j:p_{ij}^\varepsilon > 0} \log\left(\frac{(1-\delta)|S|}{\delta}p_{ij}^\varepsilon + 1\right)^2 \leq (M + d_{\text{out}})\left(\log\frac{|S|}{\delta}\right)^2$$

and

$$\begin{aligned}
L_i(\mathbf{Z}_{i*}) &= -\sum_j p_{ij}^\varepsilon \log\left(\frac{(1-\delta)|S|}{\delta}p_{ij}^\varepsilon + 1\right) + \log\sum_j\left(\frac{(1-\delta)|S|}{\delta}p_{ij}^\varepsilon + 1\right) \\
&= -\sum_j p_{ij}^\varepsilon \log\left((1-\delta)p_{ij}^\varepsilon + \frac{\delta}{|S|}\right) \\
&= -\sum_j p_{ij}^\varepsilon \log p_{ij}^\varepsilon - \sum_j p_{ij}^\varepsilon \log\left(1 - \delta + \frac{\delta}{|S|p_{ij}^\varepsilon}\right) \\
&\leq \inf L_i + O\left(\frac{\delta}{|S|\varepsilon} \vee \delta\right),
\end{aligned}$$

owing to the inequality $-\log(1+x) \leq 2|x|$ for small $x$ and the bound $p_{ij}^\varepsilon \geq c\varepsilon$ when $p_{ij}^\varepsilon \neq 0$. Moreover, from the convexity of $L_i$ and (21) we have the relation

$$\begin{aligned}
&\|\mathbf{W}_{i*}^{(t+1)} - \mathbf{Z}_{i*}\|^2 - \|\mathbf{W}_{i*}^{(t)} - \mathbf{Z}_{i*}\|^2 \\
&= -2\langle\nabla L_i(\mathbf{W}_{i*}^{(t)}), \mathbf{W}_{i*}^{(t)} - \mathbf{Z}_{i*}\rangle + \|\nabla L_i(\mathbf{W}_{i*}^{(t)})\|^2 \\
&\leq 2(L_i(\mathbf{Z}_{i*}) - L_i(\mathbf{W}_{i*}^{(t)})) + \frac{2}{\eta_0}(L_i(\mathbf{W}_{i*}^{(t)}) - L_i(\mathbf{W}_{i*}^{(t+1)})).
\end{aligned}$$

Summing over $t = 0, \cdots, T-1$ and rearranging gives

$$\begin{aligned}
L_i(\mathbf{W}_{i*}^{(T)}) &\leq \frac{1}{T}\sum_{t=0}^{T-1} L_i(\mathbf{W}_{i*}^{(t)}) \\
&\leq L_i(\mathbf{Z}_{i*}) + \frac{L_i(\mathbf{W}_{i*}^{(0)})}{\eta_0 T} + \frac{\|\mathbf{W}_{i*}^{(0)} - \mathbf{Z}_{i*}\|^2 - \|\mathbf{W}_{i*}^{(T)} - \mathbf{Z}_{i*}\|^2}{2T} \\
&\leq L_i(\mathbf{Z}_{i*}) + \frac{\log|S|}{\eta_0 T} + \frac{\|\mathbf{Z}_{i*}\|^2}{2T} \\
&\leq \inf L_i + O\left(\frac{\delta}{|S|\varepsilon} \vee \delta\right) + \frac{M + d_{\text{out}} + \eta_0^{-1}}{2T}\left(\log\frac{|S|}{\delta}\right)^2.
\end{aligned}$$

Since $|S| = O(KM)$, by taking $\delta = M/T$ if $\varepsilon \geq 1/KM$ and $\delta = KM^2\varepsilon/T$ if $\varepsilon < 1/KM$, it follows that

$$L_i(\mathbf{W}_{i*}^{(T)}) - \inf L_i = O\left(\frac{KM^2}{T}\left(\log\frac{KT}{M\varepsilon}\right)^2\right)$$

uniformly for all $i$. Again by applying (21) we obtain the bound

$$\|\nabla L_i(\mathbf{W}_{i*}^{(T)})\|^2 \leq \frac{2}{\eta_0}\left(L_i(\mathbf{W}_{i*}^{(T)}) - \mathbf{W}_{i*}^{(T+1)})\right) \leq \frac{2}{\eta_0}\left(L_i(\mathbf{W}_{i*}^{(T)}) - \inf L_i\right).$$

Since

$$\sum_j\left(\hat{p}_{\mathbf{W}^{(T)}}(e_j|e_i) - p_{ij}^\varepsilon\right)^2 = \|\nabla L_i(\mathbf{W}_{i*}^{(T)})\|^2,$$

this concludes part (1).

For part (2), by running gradient descent for time $T_1 = \widetilde{O}(KM^2\varepsilon^{-2})$ steps, the bound (18) ensures

$$\sup_{1 \leq i,j \leq S} |\hat{p}_{ij}^{(T_1)} - p_{ij}^\varepsilon| = o(\varepsilon), \tag{22}$$

and in particular $\hat{p}_{ij}^{(T_1)} = o(\varepsilon)$ if and only if $p_{ij}^\varepsilon = 0$. Hence defining the thresholded parameter matrix

$$\mathbf{W}^+ \in \mathbb{R}^{|S| \times |S|} : \quad w_{ij}^+ = \begin{cases} -\infty & \text{if } \hat{p}_{ij}^{(T_1)} < c_{\text{thres}}\varepsilon \\ w_{ij}^{(T_1)} & \text{otherwise} \end{cases}$$

the corresponding softmax scores $\hat{p}_{ij}^+ = \hat{p}_{\mathbf{W}^+}(e_j|e_i)$ are affixed to precisely zero. Moreover, note that the ratios $\hat{p}_{ij}^+ / \hat{p}_{ik}^+$ for all $j, k$ such that $\hat{p}_{ij}^+, \hat{p}_{ik}^+ > 0$ do not change before/after thresholding. Define the set $D_i = \{j \in [S] : \hat{p}_{ij}^+ > 0\}$ so that $|D_i| \le M + d_{\text{out}}$ and

$$1 - \sum_{j \in D_i} \hat{p}_{ij}^{(T_1)} = \sum_{j \in D_i} |\hat{p}_{ij}^{(T_1)} - p_{ij}^\varepsilon| \le (M + d_{\text{out}})o(\varepsilon) = o(1).$$

Therefore we have for all $j \in D_i$,

$$\hat{p}_{ij}^+ = \frac{\hat{p}_{ij}^+}{\sum_{k \in D_i} \hat{p}_{ik}^+} = \frac{\hat{p}_{ij}^{(T_1)}}{\sum_{k \in D_i} \hat{p}_{ik}^{(T_1)}} = (1 + o(1))\hat{p}_{ij}^{(T_1)}, \quad \hat{p}_{ij}^+ \ge \hat{p}_{ij}^{(T_1)},$$

and by comparing with (22) we obtain the desired bound.

Finally for part (3), we utilize the strong convexity of $L_i$ on a bounded domain. We treat all entries set to $-\infty$ in part (2) as nonexistent, so that for example $\min_j p_{ij}^\varepsilon \ge c\varepsilon > 0$. Then there exists a set of logits $\mathbf{W}^*$ such that $\hat{p}_{\mathbf{W}^*} = p_{ij}^\varepsilon$; as adding the same constant to all entries in a row does not affect the probabilities $\hat{p}_{\mathbf{W}^*}$, we may assume the row sums of $\mathbf{W}^*$ are equal to $\mathbf{W}^+$ so that $(\mathbf{W}^* - \mathbf{W}^+)\mathbf{1} = 0$.

The Hessian of $L_i$ is equal to

$$\nabla^2 L_i(\mathbf{W}_{i*}) = \operatorname{diag}\hat{p}_{i*} - \hat{p}_{i*}\hat{p}_{i*}^\top$$

and has zero curvature along the direction $\mathbf{1}$. We claim that in all orthogonal directions $\{\mathbf{1}\}^\perp$, $\nabla^2 L_i$ is $\Theta(\varepsilon^2)$-strongly convex. Indeed, for any vector $v$ such that $\|v\| = 1$ and $v^\top \mathbf{1} = 0$ and any $t \in \mathbb{R}$ we have

$$\sum_j \hat{p}_{ij}(v_j t - 1)^2 = \left( \sum_j \hat{p}_{ij}v_j^2 \right)t^2 - 2t\sum_j \hat{p}_{ij}v_j + 1 \ge \min_j \hat{p}_{ij}$$

since $v_j t \le 0$ for at least one $j$. Then the discriminant must satisfy

$$\left( \sum_j \hat{p}_{ij}v_j \right)^2 - \left( \sum_j \hat{p}_{ij}v_j^2 \right)\left( 1 - \min_j \hat{p}_{ij} \right) \le 0,$$

so that

$$v^\top \nabla^2 L_i(\mathbf{W}_{i*})v = \sum_j \hat{p}_{ij}v_j^2 - \left( \sum_j \hat{p}_{ij}v_j \right)^2 \ge \left( \sum_j \hat{p}_{ij}v_j^2 \right)\min_j \hat{p}_{ij} \ge \min_j \hat{p}_{ij}^2$$

which is $\Theta(\varepsilon^2)$ due to (19). It now follows from classical convex analysis that

$$\begin{aligned} \|\hat{p}_{i*}^{(T_1+T)} - p_{i*}^\varepsilon\| &\le \|\mathbf{W}_{i*}^{(T_1+T)} - \mathbf{W}_{i*}^*\| \\ &\le (1 - \Omega(\varepsilon^2))^T\|\mathbf{W}_{i*}^+ - \mathbf{W}_{i*}^*\| \\ &= \exp(-\Omega(\varepsilon^2 T)) \cdot O(\log \varepsilon^{-1}), \end{aligned}$$

where we have again used that softmax is 1-Lipschitz and the effective learning rate $\mu_i \eta = \Theta(1)$. $\qquad \square$

## D.2. Analysis of Search and PPO

We first show that the initial cluster exploration phase of the search algorithm is consistent.

**Lemma D.2.** *For each outer loop of Algorithm 2, after $T_0$ steps, $\hat{C}$ returns the cluster $C_k$ containing $X_0$ with probability $1 - \widetilde{O}(K^{-2})$.*

*Proof.* $\hat{C} \neq C_k$ implies that either some state $y \in C_k$ has not been hit by some $X_t^{n,\varepsilon}$ by time $T_0$, or all chains $X_t^{n,\varepsilon}$ have reached some point outside $C_k$ at time $T_0$.

Denote the hitting time of $C \subseteq S$ by $X_t^{n,\varepsilon}$ as $\tau_C^{n,\varepsilon}$. Since $K \leq \mathrm{poly}(M)$, by Lemma B.9 it holds that

$$\mathbb{P}_{X_0}(\tau_y^{n,\varepsilon} \geq T_0) \leq \frac{1}{MK^2}, \quad \forall y \in C_k$$

by choosing $T_0 \geq \Omega(M(\log M)^2)$. Union bounding over $y, n$ gives

$$\mathbb{P}_{X_0}\left(\max_{y \in C_k} \max_{n \leq N} \tau_y^{n,\varepsilon} > T_0\right) \leq MN \cdot \mathbb{P}_{X_0}(\tau_y^{1,\varepsilon} > T_0) \leq O\left(\frac{\log K}{K^2}\right).$$

Moreover by the argument in Lemma C.12, since each $X^{n,\varepsilon}$ is independently generated from $p^\varepsilon$,

$$\mathbb{P}_{X_0}\left(\max_{n \leq N} \tau_{C_k^c}^{n,\varepsilon} \leq T_0\right) \leq \mathbb{P}_{X_0}\left(\tau_{C_k^c}^{1,\varepsilon} \leq T_0\right)^N$$

$$\leq \left(1 - \left(1 - \Theta\left(\frac{\varepsilon}{M}\right)\right)^{T_0}\right)^N$$

$$\leq \exp\left(-N \exp\left(-\Theta\left(\frac{T_0 \varepsilon}{M}\right)\right)\right)$$

$$\leq \exp(-N/2) \leq K^{-2}$$

by ensuring that $N \geq 4\log K$. Hence we have shown that $\mathbb{P}_{X_0}(\hat{C} \neq C_k) \leq \widetilde{O}(K^{-2})$. $\qquad\square$

**Proposition D.3** (Proposition 3.3 restated). *PRM mode of Algorithm 2 returns $\mathcal{M}_s = E_s$ with probability $1 - \widetilde{O}(1/K)$.*

*Proof.* Denote the set of outbound edges from $C_k$ as $E_{s,k} := \{(x,y) \in E_s : x \in C_k, y \in C_k^c\}$ so that $|E_{s,k}| \leq n_{\mathrm{out}} d_{\mathrm{out}}$ and fix $(x,y) \in E_{s,k}$. The probability that a fixed rollout $X^{1,\varepsilon}$ takes the edge $(x,y)$ and terminates within time $T_{\max}$ is bounded below as

$$\mathbb{P}_{X_0}\left(\exists t \leq T_{\max} : X_{t-1}^{1,\varepsilon} = x, X_t^{1,\varepsilon} = y\right)$$

$$= \sum_{t=1}^{T_{\max}} \mathbb{P}_{X_0}\left(\tau_{C_k^c}^{1,\varepsilon} \geq t, X_{t-1}^{1,\varepsilon} = x, X_t^{1,\varepsilon} = y\right)$$

$$\geq \sum_{t=1}^{T_{\max}} \mathbb{P}_{X_0}(\tilde{X}_{t-1}^{k,\varepsilon} = x) \mathbb{P}_{X_0}(\tau_{C_k^c}^{1,\varepsilon} \geq t) p^\varepsilon(y|x)$$

$$\geq \sum_{t=2t_{\mathrm{mix}}}^{T_{\max}} \frac{\rho}{2M}\left(1 - \Theta\left(\frac{\varepsilon}{M}\right)\right)^{t-t_{\mathrm{mix}}} \Theta(\varepsilon)$$

$$\geq \left(1 - \Theta\left(\frac{\varepsilon}{M}\right)\right)^{t_{\mathrm{mix}}}\left(1 - \left(1 - \Theta\left(\frac{\varepsilon}{M}\right)\right)^{T_{\max}-2t_{\mathrm{mix}}+1}\right)$$

$$\geq \exp\left(-O\left(\frac{\varepsilon \log M}{M}\right)\right) \cdot \exp\left(-\exp\left(-\Theta\left(\frac{T_{\max}\varepsilon}{M}\right)\right)\right) \geq c,$$

for some positive constant $c$. Therefore by union bounding over all edges in $E_{s,k}$,

$$\mathbb{P}_{X_0}(\hat{E} \neq E_{s,k}) \leq \sum_{(x,y) \in E_{s,k}} \mathbb{P}_{X_0}\left(\nexists t \leq T_{\max} : X_{t-1}^{1,\varepsilon} = x, X_t^{1,\varepsilon} = y\right)^N$$

$$\leq |E_{s,k}|(1-c)^N \leq O(1/K^2)$$

by taking $N/\log K$ suitably large. By union bounding again over all $R = \Theta(K \log K)$ iterations of the outer loop, the probability that the inner loop fails to return the correct set of sparse edges for some iteration is at most $\widetilde{O}(1/K)$. Finally, the probability that some cluster will be missed during the $R$ iterations is bounded above for a suitable choice of $R$ as

$$K \left(1 - \Theta(1/K)\right)^R \leq K \exp\left(-\Theta(R/K)\right) \leq O(1/K).$$

Thus with probability $1 - \widetilde{O}(1/K) - O(1/K)$, all clusters $C_k$ will be explored and the correct set of sparse edges $E_{s,k}$ will always be added to $\mathcal{M}_s$, so that the final output satisfies $\mathcal{M}_s = E_s$. $\qquad\square$

We now prove the convergence of the PPO-Clip algorithm.

*Proof.* As in the proof of Proposition 3.3, we condition on the event that the correct set of sparse edges is returned for all clusters, and focus on a single cluster $C_k$. Again write $\hat{p}_{ij}^{(t)} = \hat{p}_{\mathbf{W}^{(t)}}(e_j|e_i)$, let $X_0 \sim \mu$ where $\mu_i = \Theta(KM)$ and denote $D_{s,i} = \{j \in [S] : (i,j) \in E_s\} \subset D_i$. The objective $L_{\text{PPO}}$ reduces to $\sum_{i \in C_k} \mu_i L_i$ where

$$L_i(\mathbf{W}; \hat{A}) = \sum_j p_{ij}^\varepsilon \min \left\{ \frac{\hat{p}_{\mathbf{W}}(e_j|e_i)}{p_{ij}^\varepsilon}, c_{\text{clip}} \right\} \hat{A}(e_i, e_j) = \sum_{j \in D_{s,i}} \min\{\hat{p}_{\mathbf{W}}(e_j|e_i), c_{\text{clip}} \cdot p_{ij}^\varepsilon\}.$$

We will show inductively that either $\hat{p}_{\mathbf{W}}(e_j|e_i) < c_{\text{clip}} \cdot p_{ij}^\varepsilon$ for all $j \in D_{s,i}$ or $\hat{p}_{\mathbf{W}}(e_j|e_i) \geq c_{\text{clip}} \cdot p_{ij}^\varepsilon$ for all $j \in D_{s,i}$ while running Algorithm 3. Assuming the former, we see that for all $j \in [S]$,

$$\frac{\mathrm{d}L_i}{\mathrm{d}w_{ij}}(\mathbf{W}^{(t)}; \hat{A}) = \frac{\mathrm{d}}{\mathrm{d}w_{ij}} \sum_{k \in D_{s,i}} \hat{p}_{ik}^{(t)} = \frac{\mathrm{d}}{\mathrm{d}w_{ij}} \sum_{k \in D_{s,i}} \frac{\exp w_{ik}^{(t)}}{\sum_\ell \exp w_{i\ell}^{(t)}}$$

$$= 1_{\{j \in D_{s,i}\}} \frac{\exp w_{ij}^{(t)}}{\sum_\ell \exp w_{i\ell}^{(t)}} - \sum_{k \in D_{s,i}} \frac{\exp w_{ik}^{(t)} \cdot \exp w_{ij}^{(t)}}{(\sum_\ell \exp w_{i\ell}^{(t)})^2}$$

$$= \hat{p}_{ij}^{(t)} \left( 1_{\{j \in D_{s,i}\}} - \sum_{k \in D_{s,i}} \hat{p}_{ik}^{(t)} \right).$$

This implies that $\frac{\mathrm{d}L_i}{\mathrm{d}w_{ij}}(\mathbf{W}^{(t)}, \hat{A}) > 0$ if and only if $j \in D_{s,i}$, so sign gradient ascent gives for each $i$ the update rule

$$w_{ij}^{(t+1)} = \begin{cases} w_{ij}^{(t)} + \mu_i \alpha & j \in D_{s,i} \\ w_{ij}^{(t)} - \mu_i \alpha & j \notin D_{s,i}. \end{cases}$$

In particular, the relative magnitudes of $\hat{p}_{ij}^{(t)}$ for all $j \in D_{s,i}$ are preserved starting from $\hat{p}_{ij}^{(0)} = p_{ij}^\varepsilon$, so that the earlier assertion is justified. Furthermore since $\hat{p}_{ij}^{(t)} \propto \exp w_{ij}^{(t)}$ for all $j$, we can derive $\hat{p}_{ij}^{(t)}$ from $\hat{p}_{ij}^{(0)}$ by directly multiplying $e^{\pm \mu_i \alpha}$ (or $e^{2\mu_i \alpha}$ and 1) and normalizing afterwards,

$$\tilde{p}_{ij}^{(t)} := \begin{cases} e^{2\mu_i \alpha t} p_{ij}^\varepsilon & j \in D_{s,i} \\ p_{ij}^\varepsilon & j \notin D_{s,i} \end{cases} \quad \text{and} \quad \hat{p}_{ij}^{(t)} = \frac{\tilde{p}_{ij}^{(t)}}{\sum_k \tilde{p}_{ik}^{(t)}}.$$

By choosing $c_{\text{clip}} = (1 - o(1))\varepsilon_{\max}/\varepsilon$ so that

$$T_i = \frac{1}{2\mu_i \alpha} \log(1 + o(1))c_{\text{clip}} = \frac{1}{2\mu_i \alpha} \log \frac{\varepsilon_{\max}}{\varepsilon},$$

we ensure that gradient ascent has not yet terminated (reached the clip threshold) by time $T_i$ and also $\tilde{p}_{ij}^{(t)} \leq \varepsilon_{\max}$ for all $j \in D_{s,i}$. This implies

$$1 \leq \sum_j \tilde{p}_{ij}^{(t)} \leq 1 + (e^{2\mu_i \alpha t} - 1) \sum_{j \in D_{s,i}} p_{ij}^\varepsilon \leq 1 + (e^{2\mu_i \alpha t} - 1)|D_{s,i}|\varepsilon \leq 1 + O(\varepsilon_{\max}).$$

Thus $\hat{p}_{ij}^{T_i} = (1 - o(1))\frac{\varepsilon_{\max}}{\varepsilon}p_{ij}^\varepsilon = p_{ij}^{\varepsilon'}$ where $\varepsilon' = (1 - o(1))\varepsilon_{\max}$ for $j \in D_{s,i}$. Moreover $\hat{p}_{ij}^{T_i}$ for all $j \in D_{s,i}$ decrease proportionally from $p_{ij}^\varepsilon$ so that they also equal the corresponding $p_{ij}^{\varepsilon'}$ values by Assumption 2.

Now if we take $T_{\text{PPO}} = \max_i T_i$, then this will hold for all states $e_i$ in the initialized cluster $C_k$ as each row will stop updating after time $T_i$. Finally, since the values $\hat{p}_{ij}$ for $e_i$ in a different cluster do not affect the above derivation, we may repeat this argument for all explored clusters to obtain the guarantee. We remark that exploring the same cluster multiple times does not affect the outcome since the clipping operation will prevent the weights from being updated from the second time onwards. $\square$

### D.3. Analysis of Distillation

**Lemma D.4.** $(Y_0, Y_1) \sim D_{\text{dist}}$ *is distributed as* $Y_0 \sim \pi^\varepsilon$, $Y_1 \sim q_{\circ}^\varepsilon(\cdot|Y_0)$ *where*

$$q_{\circ}^\varepsilon(x_\ell|x_k) := \pi_k^\varepsilon(x_k)\mathbb{P}_{x_k}(X_{\bar{\tau}_{S_\circ}^\varepsilon}^\varepsilon = x_\ell), \quad k \neq \ell,$$

$$q_{\circ}^\varepsilon(x_k|x_k) := 1 - \sum_{\ell \neq k} q_{\circ}^\varepsilon(x_\ell|x_k).$$

*Proof.* The $Y_0$ component of $D_{\text{dist}}$ is clearly distributed according to $\pi^\varepsilon$. Consider samples such that $X_t^\varepsilon \in C_k$, in which case $Y_0^{(t)} = x_k$ and $X_t^\varepsilon \sim \pi_k^\varepsilon$. If $X_t^\varepsilon \neq x_k$, then $Y_0^{(t)} = Y_1^{(t)} = x_k$. If $X_t^\varepsilon = x_k$, the following state $Y_1^{(t)}$ is the next return of $X^\varepsilon$ to $S_\circ$, so that

$$\mathbb{P}_{Y_0^{(t)}}(Y_1^{(t)} = x_\ell) = \mathbb{P}_{Y_0^{(t)}}(Y_1^{(t)} = x_\ell, X_t^\varepsilon = x_k) = \pi_k^\varepsilon(x_k)\mathbb{P}_{x_k}(X_{\bar{\tau}_{S_\circ}^\varepsilon}^\varepsilon = x_\ell)$$

holds for all $\ell \neq k$. Comparing with the definition of $q_{\circ}^\varepsilon$, this shows that $Y_1 \sim q_{\circ}^\varepsilon(\cdot|Y_0)$ for $(Y_0, Y_1) \sim D_{\text{dist}}$. $\square$

**Proposition D.5** (Proposition 4.1 restated). *Denote the return time of $q_{\circ}^\varepsilon$ to $x_k$ as $\bar{\tau}_{\circ,x_k}^\varepsilon$. For all $k, \ell \in [K]$ with $k \neq \ell$, it holds that*

$$\frac{\mathbb{P}_{x_k}(\bar{\tau}_{\circ,x_\ell}^\varepsilon < \bar{\tau}_{\circ,x_k}^\varepsilon)}{q_{\star}^\varepsilon(C_\ell|C_k)} = 1 + \widetilde{O}\left(\frac{1}{\log M}\right).$$

We remark that the asymptotic version of this result is Theorem 5.3 of Betz & Le Roux (2016). It is also shown that this is the best characterization of the metastable dynamics, as the transition probabilities $q_{\circ}^\varepsilon$ themselves cannot be made to be independent of the choice of representatives $S_\circ$ even in the asymptotic limit.

*Proof.* Applying Corollary C.7 and Proposition C.10 to the representatives $x_k, x_\ell$ of $C_k, C_\ell$ gives

$$\frac{\pi_k^\varepsilon(x_k)\mathbb{P}_{x_k}(\bar{\tau}_{x_\ell}^\varepsilon < \bar{\tau}_{x_k}^\varepsilon)}{q_{\star}^\varepsilon(C_\ell|C_k)} = 1 + \widetilde{O}\left(\frac{1}{\log M}\right).$$

Considering the numerator, conditioning the event $\{\bar{\tau}_{x_\ell}^\varepsilon < \bar{\tau}_{x_k}^\varepsilon\}$ on the first return of $X^\varepsilon$ to $S_\circ$, we have that

$$\mathbb{P}_{x_k}(\bar{\tau}_{x_\ell}^\varepsilon < \bar{\tau}_{x_k}^\varepsilon) = \mathbb{P}_{x_k}(X_{\bar{\tau}_{S_\circ}^\varepsilon}^\varepsilon = x_\ell) + \sum_{m \neq k, \ell} \mathbb{P}_{x_k}(X_{\bar{\tau}_{S_\circ}^\varepsilon}^\varepsilon = x_m)\mathbb{P}_{x_m}(\bar{\tau}_{x_\ell}^\varepsilon < \bar{\tau}_{x_k}^\varepsilon)$$

and multiplying both sides by $\pi_k^\varepsilon(x_k)$ gives

$$\pi_k^\varepsilon(x_k)\mathbb{P}_{x_k}(\bar{\tau}_{x_\ell}^\varepsilon < \bar{\tau}_{x_k}^\varepsilon) = q_{\circ}^\varepsilon(x_\ell|x_k) + \sum_{m \neq k, \ell} q_{\circ}^\varepsilon(x_m|x_k)\mathbb{P}_{x_m}(\bar{\tau}_{x_\ell}^\varepsilon < \bar{\tau}_{x_k}^\varepsilon).$$

Now note that $q_{\circ}^\varepsilon$ is a rescaled version of $X^\varepsilon$ reduced to $S_\circ$ where only the diagonal elements $q_{\circ}^\varepsilon(x_k|x_k)$ have been increased and all other elements have been decreased proportionally. It follows that $\mathbb{P}_{x_m}(\bar{\tau}_{x_\ell}^\varepsilon < \bar{\tau}_{x_k}^\varepsilon) = \mathbb{P}_{x_m}(\bar{\tau}_{\circ,x_\ell}^\varepsilon < \bar{\tau}_{\circ,x_k}^\varepsilon)$ for all $m \neq k, \ell$. Therefore applying the same argument to the chain $q_{\circ}^\varepsilon$ we obtain

$$\pi_k^\varepsilon(x_k)\mathbb{P}_{x_k}(\bar{\tau}_{x_\ell}^\varepsilon < \bar{\tau}_{x_k}^\varepsilon) = q_{\circ}^\varepsilon(x_\ell|x_k) + \sum_{m \neq k, \ell} q_{\circ}^\varepsilon(x_m|x_k)\mathbb{P}_{x_m}(\bar{\tau}_{\circ,x_\ell}^\varepsilon < \bar{\tau}_{\circ,x_k}^\varepsilon)$$

$$= \mathbb{P}_{x_k}(\bar{\tau}_{\circ,x_\ell}^\varepsilon < \bar{\tau}_{\circ,x_k}^\varepsilon),$$

concluding the proof. $\square$

For the convergence of the distilled model $\hat{q}_{\mathbf{Z}}$, we first show the following.

**Lemma D.6.** *Under Assumption 5, for any $k \neq \ell$, $q_\circ^\varepsilon(x_\ell | x_k) = \Theta(\varepsilon/M)$ if there is a sparse edge from $C_k$ to $C_\ell$ and $q_\circ^\varepsilon(x_\ell | x_k) = 0$ otherwise.*

*Proof.* If there is no sparse edge from $C_k$ to $C_\ell$, the first return of $X^\varepsilon$ to $S_\circ$ starting from $x_k$ cannot be $x_\ell$, since $X^\varepsilon$ must first travel to a different cluster $C_n \neq C_\ell$ to escape $C_k$ where it will inevitably hit $x_n$. Thus $\mathbb{P}_{x_k}(X^\varepsilon_{\bar{\tau}^\varepsilon_{S_\circ}} = x_\ell) = 0$ and $q_\circ^\varepsilon(x_\ell | x_k) = 0$.

Suppose there exists a sparse edge $(x', x_\ell)$ where $x' \in C_k$. Note that

$$q_\circ^\varepsilon(x_\ell | x_k) \leq \pi_k^\varepsilon(x_k) \mathbb{P}_{x_k}(\bar{\tau}^\varepsilon_{x_\ell} < \bar{\tau}^\varepsilon_{x_k}) = q_\star^\varepsilon(C_\ell | C_k)\left(1 + \widetilde{O}\left(\frac{1}{\log M}\right)\right) = O(\varepsilon/M)$$

by Corollary C.7, Proposition C.10 and Assumption 4. Moreover letting $\Gamma_{x_k, x'}$ be a simple path with positive $p^\varepsilon$-probability in $C_k$, it holds that

$$q_\circ^\varepsilon(x_\ell | x_k) \geq \pi_k^\varepsilon(x_k) \mathbb{P}_{x_k}(\Gamma_{x_k, x'}) p^\varepsilon(x', x_\ell) \geq \Omega(\varepsilon/M),$$

proving the assertion. $\qquad\square$

The proof of Proposition 4.2 then follows by repeating the analysis of Theorem D.1, replacing dimension $|S|$ by $K$, the maximum number of outgoing edges $M + d_{\text{out}}$ by the number of sparse edges $d_{\text{out}}$, and the probability lower threshold $\Theta(\varepsilon)$ by $\Theta(\varepsilon/M)$. The learning rate $\eta = \Theta(K)$ is justified since $\mathbb{P}(Y_0 = x_k) = \pi^\varepsilon(C_k) = \Theta(1/K)$ by Corollary C.9. This results in a convergence rate of $O(\log KT/T)$ for the initial stage and $\exp(-\Omega(\varepsilon^2 T/M^2))$ after thresholding. The details are omitted. $\qquad\square$

We also require the following dynamical properties of $q_\circ^\varepsilon$:

**Lemma D.7.** *The stationary distribution $\pi_\circ^\varepsilon$ of $q_\circ^\varepsilon$ on $S_\circ$ satisfies $\pi_\circ^\varepsilon(x_k) = \Theta(1/K)$, moreover, $\mathbb{E}_{x_k}[\tau^\varepsilon_{\circ, x_\ell}] = O(KM/\varepsilon)$ for all $k, \ell$.*

*Proof.* By Proposition B.10 and Proposition 4.1, we have that

$$\frac{\pi_\circ^\varepsilon(x_k)}{\pi_\circ^\varepsilon(x_\ell)} = \frac{\mathbb{P}_{x_\ell}(\bar{\tau}^\varepsilon_{\circ, x_k} < \bar{\tau}^\varepsilon_{\circ, x_\ell})}{\mathbb{P}_{x_k}(\bar{\tau}^\varepsilon_{\circ, x_\ell} < \bar{\tau}^\varepsilon_{\circ, x_k})} = \frac{q_\star^\varepsilon(C_k | C_\ell)}{q_\star^\varepsilon(C_\ell | C_k)}\left(1 + \widetilde{O}\left(\frac{1}{\log M}\right)\right) = \Theta(1)$$

due to Assumption 4. Then each $\pi_\circ^\varepsilon(x_k)$ must be of order $1/K$. It further follows from an application of (8) that

$$\mathbb{E}_{x_k}[\tau^\varepsilon_{\circ, x_\ell}] \leq \frac{1}{\pi_\circ^\varepsilon(x_k)\mathbb{P}_{x_k}(\bar{\tau}^\varepsilon_{\circ, x_\ell} < \bar{\tau}^\varepsilon_{\circ, x_k})} = \Theta\left(\frac{KM}{\varepsilon}\right).$$

$\qquad\square$

**Theorem D.8** (Theorem 4.3 restated). *For all $k \neq \ell$, $\hat{q}_{\mathbf{Z}^+}(x_\ell | x_k) = \Theta(1)$ if there exists a sparse edge from $C_k$ to $C_\ell$ or 0 if not. Moreover, the hitting time $\tau^+_{x_\ell}$ of $x_\ell \in S_\circ$ by $\hat{q}_{\mathbf{Z}^+}$ satisfies $\mathbb{E}_{x_k}[\tau^+_{x_\ell}] = O(K)$.*

*Proof.* The chain $q_\circ^\varepsilon$ can be retrieved from $\hat{q}_{\mathbf{Z}^+}$ by making it 'lazy.' Indeed, note that $\hat{q}_{\mathbf{Z}^+}$ is computed from the distilled model $q_\circ^\varepsilon$ as

$$\hat{q}_{\mathbf{Z}^+}(x_\ell | x_k) = \begin{cases} \dfrac{e^\beta q_\circ^\varepsilon(x_\ell | x_k)}{q_\circ^\varepsilon(x_k | x_k) + e^\beta \sum_{\ell' \neq k} q_\circ^\varepsilon(x_\ell | x_k)} & \ell \neq k, \\[4mm] \dfrac{q_\circ^\varepsilon(x_k | x_k)}{q_\circ^\varepsilon(x_k | x_k) + e^\beta \sum_{\ell' \neq k} q_\circ^\varepsilon(x_\ell | x_k)} & \ell = k. \end{cases}$$

Since the sum in the denominator is over at most $d_{\text{out}} + 1$ nonzero terms, by choosing $e^\beta = \Theta(M/\varepsilon)$ we ensure that $\hat{q}_{\mathbf{Z}^+}(x_\ell | x_k) = \Theta(1)$ for those terms that are nonzero. Conversely, by viewing the logits of $q_\circ^\varepsilon$ as obtained by subtracting $\beta$

from $\mathbf{Z}^+$, we have

$$q_\circ^\varepsilon(x_\ell|x_k) = \begin{cases} \dfrac{e^{-\beta}\hat{q}_{\mathbf{Z}^+}(x_\ell|x_k)}{\hat{q}_{\mathbf{Z}^+}(x_k|x_k) + e^{-\beta}\sum_{\ell'\neq k}\hat{q}_{\mathbf{Z}^+}(x_\ell|x_k)} & \ell \neq k, \\[3ex] \dfrac{\hat{q}_{\mathbf{Z}^+}(x_k|x_k)}{\hat{q}_{\mathbf{Z}^+}(x_k|x_k) + e^{-\beta}\sum_{\ell'\neq k}\hat{q}_{\mathbf{Z}^+}(x_\ell|x_k)} & \ell = k. \end{cases}$$

This may be expressed as

$$q_\circ^\varepsilon(x_\ell|x_k) = \lambda_k\hat{q}_{\mathbf{Z}^+}(x_\ell|x_k) + (1-\lambda_k)\delta_{k\ell}$$

where

$$\lambda_k = \frac{1}{e^\beta\hat{q}_{\mathbf{Z}^+}(x_k|x_k) + \sum_{\ell'\neq k}\hat{q}_{\mathbf{Z}^+}(x_\ell|x_k)} = e^{-\beta}q_\circ^\varepsilon(x_k|x_k) + \sum_{\ell'\neq k}q_\circ^\varepsilon(x_\ell|x_k) = O(\varepsilon/M).$$

Hence $q_\circ^\varepsilon$ is equivalent to the lazy chain obtained from $\hat{q}_{\mathbf{Z}}^+$ by inserting additional self-transitions of each $x_k$ with probability $1-\lambda_k$. It follows that the expected hitting time $\mathbb{E}_{x_k}[\tau_{\circ,x_\ell}^\varepsilon]$ is larger than $\mathbb{E}_{x_k}[\tau_{x_\ell}^+]$ by at least a factor of $\min_k \lambda_k^{-1} = \Omega(M/\varepsilon)$. Comparing with Lemma D.7 proves the assertion. □

# E. Proofs for Hardness Results

We first review the definition of a group action.

**Definition E.1** (group action). Let $(G,\circ)$ be a group with identity $e_G$ and $\mathcal{R}$ be any set. $G$ is said to *act* on $\mathcal{R}$ (from the left) if there exists a map $\cdot : G \times \mathcal{R} \to \mathcal{R}$ (the *group action*) satisfying the following two axioms:

(1) (identity) $e_G \cdot r = r$ for all $r \in \mathcal{R}$,

(2) (composition) $g_1 \cdot (g_2 \cdot r) = (g_1 \circ g_2) \cdot r$ for all $g_1, g_2 \in G$ and $r \in \mathcal{R}$.

It follows that the map $r \mapsto g \cdot r$ is a bijection for all $g \in G$, with inverse $r \mapsto g^{-1} \cdot r$.

## E.1. Proof of Theorem 5.2

Let $\mathcal{P}' \subseteq \mathcal{P}$ be such that $|\mathcal{P}'| = \text{SDA}(\mathcal{P}, \mathcal{I})$, $h_p$ are pairwise orthogonal and $\mathcal{I}_p$ are equal for all $p \in \mathcal{P}'$. This ensures that $f_\theta(x, \mathcal{I}_p(x))$ is independent of $p$. Choosing $p$ uniformly randomly from $\mathcal{P}'$, the variance of the gradient of $L$ with respect to $p$ is computed as

$$\begin{aligned} \text{Var}_p \nabla L(\theta;p) &= \min_{u\in\mathbb{R}} \mathbb{E}_p\left[(-2\langle\nabla f_\theta, h_p - f_\theta\rangle_{\mathcal{H}} - u)^2\right] \\ &\leq \mathbb{E}_p\left[4\langle\nabla f_\theta, h_p\rangle_{\mathcal{H}}^2\right] \\ &= \frac{4}{|\mathcal{P}'|}\sum_{p\in\mathcal{P}'}\langle\nabla f_\theta, h_p\rangle_{\mathcal{H}}^2 \\ &\leq \frac{4\|\nabla f_\theta\|_{\mathcal{H}}^2}{|\mathcal{P}'|} \leq \delta^3. \end{aligned}$$

It follows from Chebyshev's inequality that for all $\theta$,

$$\mathbb{P}\left(\|\nabla L(\theta;p) - \mathbb{E}_p[\nabla L(\theta;p)]\| \geq \delta\right) \leq \delta.$$

Thus if all queries to $\nabla L$ are adversarially corrupted with $\delta$ to return $\mathbb{E}_p[\nabla L(\theta;p)]$ when $\|\nabla L(\theta;p) - \mathbb{E}_p[\nabla L(\theta;p)]\| < \delta$, the $n$ successive queries will not reveal any information on the ground truth $p$ with probability $1-n\delta$ by a union bound. Hence after running the algorithm, the loss is bounded below by the random guessing error with any fixed $\theta$. Noting that replacing $f_\theta$ with $\bar{f}_\theta(x) := \max\{\min\{f_\theta(x),1\},-1\}$ does not increase the loss, it follows that

$$\begin{aligned} \mathbb{E}_p[L(\theta;p)] &\geq \mathbb{E}_p\left[\|h_p - \bar{f}_\theta\|_{\mathcal{H}}^2\right] \\ &= 1 + \|\bar{f}_\theta\|_{\mathcal{H}}^2 - 2\mathbb{E}_p\left[\langle h_p, \bar{f}_\theta\rangle_{\mathcal{H}}\right] \end{aligned}$$

$$= 1 + \|\bar{f}_\theta\|_{\mathcal{H}}^2 - \frac{2}{|\mathcal{P}'|} \left\langle \sum_{p \in \mathcal{P}'} h_p, \bar{f}_\theta \right\rangle_{\mathcal{H}}$$

$$= 1 + \left\| \bar{f}_\theta - \frac{1}{|\mathcal{P}'|} \sum_{p \in \mathcal{P}'} h_p \right\|_{\mathcal{H}}^2 - \frac{1}{|\mathcal{P}'|^2} \sum_{p,p' \in \mathcal{P}'} \langle h_p, h_{p'} \rangle_{\mathcal{H}}$$

$$\geq 1 - \frac{1}{|\mathcal{P}'|},$$

uniformly for all $\theta$. $\qquad\square$

### E.2. Proof of Theorem 5.3

Note that $\mathrm{SDA}(\mathcal{P}; \mathcal{P}) = 1$ is trivial. We give three constructions realizing the lower bounds in order of strictness of the additional information constraint.

**Lower bounding** $\mathrm{SDA}(\mathcal{P}; \varnothing)$. Suppose $p = p^\varepsilon$ is any kernel satisfying Assumptions 1, 2 and $\mathcal{D}$ is any distribution satisfying Assumption 3'. We first analyze the case where the lower bound of Assumption 3' is $\Omega(\log K)$ and $M \geq \Omega(K)$, since the proof is slightly more involved. Let $s_{k,1}, \cdots, s_{k,|C_k|}$ be a labeling of all states in $C_k$ such that all nodes with outbound sparse edges are contained in the first $n_{\mathrm{out}}$ nodes. Define the quantity

$$q := \left\lfloor \frac{\min_{k \in K} |C_k|}{n_{\mathrm{out}}} \right\rfloor = \Theta(M).$$

For a vector $v \in \mathbb{Z}_q^K$, define the corresponding permutation of $S$ (also denoted by $v$) as

$$v(s_{k,i}) = \begin{cases} s_{k,(i+v_k n_{\mathrm{out}}-1 \mod q n_{\mathrm{out}})+1} & 1 \leq i \leq q n_{\mathrm{out}} \\ s_{k,i} & q n_{\mathrm{out}} < i \leq |C_k|. \end{cases}$$

That is, the first $q n_{\mathrm{out}}$ states of $C_k$ are cyclically shifted by $v_k$-multiples of $n_{\mathrm{out}}$ and the remaining states are left fixed. Denote the pushforward kernel of $p^\varepsilon$ induced by $v$ as

$$v_\sharp p^\varepsilon(y|x) = p^\varepsilon(v^{-1}(y)|v^{-1}(x)).$$

It is clear that $v_\sharp p^\varepsilon \in \mathcal{P}$ for all $v$, moreover, Assumptions 1, 2 hold when replacing $E_s$ by $E_s(v_\sharp p^\varepsilon)$ since $v$ only permutes states within clusters and does not affect the sparse structure.

**Lemma E.2.** *Let $d_H$ denote the Hamming distance on $\mathbb{Z}_q^K$. For any two $v, v' \in \mathbb{Z}_q^K$ it holds that*

$$|E_s(v_\sharp p^\varepsilon)| \leq n_{\mathrm{out}} d_{\mathrm{out}} K$$

*and*

$$\left| E_s(v_\sharp p^\varepsilon) \cap E_s(v'_\sharp p^\varepsilon) \right| \leq n_{\mathrm{out}} d_{\mathrm{out}} (K - d_H(v, v')).$$

*Proof.* Suppose for some $x \in C_k, y \in C_\ell$ with $k \neq \ell$ we have $(x, y) \in E_s(v_\sharp p^\varepsilon)$. Then $(v^{-1}(x), v^{-1}(y)) \in E_s(p^\varepsilon)$ so that $v^{-1}(x) = s_{k,i}$ for some $1 \leq i \leq n_{\mathrm{out}}$, hence $x = v(s_{k,i}) = s_{k,i+v_k n_{\mathrm{out}}}$. If at the same time $(x, y) \in E_s(v'_\sharp p^\varepsilon)$, it must hold that $s_{k,i+v_k n_{\mathrm{out}}} = s_{k,i+v'_k n_{\mathrm{out}}}$ and so $k$ must satisfy $v_k = v'_k$. There are exactly $K - d_H(v, v')$ such clusters and at most $n_{\mathrm{out}} d_{\mathrm{out}}$ sparse edges leading out of each cluster, concluding the second bound. The first bound follows from the second by setting $v = v'$. $\qquad\square$

We construct a well-separated subset of $\mathbb{Z}_q^K$ via the Gilbert-Varshamov bound.

**Lemma E.3** (Gilbert-Varshamov bound). *The maximum size $A_q(K, d)$ of a code of length $K$ over an alphabet of size $q$ with minimum Hamming distance $d$ satisfies*

$$A_q(K, d) \geq \frac{q^K}{\mathrm{Vol}_q(K, d-1)}, \quad \mathrm{Vol}_q(K, d) = \sum_{i=0}^{d} \binom{K}{i} (q-1)^i.$$

*Moreover for $\tau \in [0, 1 - 1/q]$ it holds that*

$$\mathrm{Vol}_q(K, \tau K) \le q^{H_q(\tau)K},$$

*where $H_q$ is the q-ary entropy function*

$$H_q(\tau) = \tau \log_q(q - 1) - \tau \log_q \tau - (1 - \tau) \log_q(1 - \tau).$$

While the classical bound only guarantees the existence of large subsets with linear (at least $K/q$) overlapping bits, we can obtain a better separation by scaling $q$ (equivalently, $M$) along with $K$. In particular, choosing the relative overlap as $O(\log q/q)$ (note the use of the natural logarithm), we obtain:

**Lemma E.4.** *For $C > 0$, $\tau = 1 - \log q/(Cq)$ and sufficiently large $q$, it holds that*

$$A_q(K, \tau K) \ge q^{\frac{\log \log q}{2Cq} K}. \tag{23}$$

*Proof.* Using the inequality $0 \le x - \log(1 + x) \le x^2$ for $|x| \le 1/2$, we may bound

$$
\begin{aligned}
H_q(\tau) &= \left(1 - \frac{\log q}{Cq}\right) \log_q(q - 1) - \left(1 - \frac{\log q}{Cq}\right) \log_q\left(1 - \frac{\log q}{Cq}\right) - \frac{\log q}{Cq} \log_q \frac{\log q}{Cq} \\
&= 1 - \frac{\log \log q}{Cq} + \frac{\log C}{Cq} + \left(\frac{1}{\log q} - \frac{1}{Cq}\right) \left(\log\left(1 - \frac{1}{q}\right) - \log\left(1 - \frac{\log q}{Cq}\right)\right) \\
&\le 1 - \frac{\log \log q}{Cq} + \frac{\log C}{Cq} + \frac{1}{\log q}\left(-\frac{1}{q} + \frac{\log q}{Cq} + \frac{(\log q)^2}{C^2 q^2}\right) \\
&\le 1 - \frac{\log \log q}{2Cq}
\end{aligned}
$$

for sufficiently large $q$. The statement follows from Lemma E.3. $\qquad\square$

Now denote by $\mathcal{V} \subset \mathbb{Z}_q^K$ the $\tau K$-separated subset of size $A_q(K, \tau K)$. For any distinct $v, v' \in \mathcal{V}$, by Lemma E.2 we have

$$\left|E_s(v_\sharp p^\varepsilon) \cap E_s(v'_\sharp p^\varepsilon)\right| \le n_{\mathrm{out}} d_{\mathrm{out}}(K - \tau K) = n_{\mathrm{out}} d_{\mathrm{out}}\left(\frac{\log q}{Cq}\right) K.$$

On the other hand, it holds that

$$\left|\mathcal{M}_{v_\sharp p^\varepsilon}(X_{\mathrm{in}}, X_{\mathrm{out}}) \cap E_s(v_\sharp p^\varepsilon)\right| = \left|\mathcal{M}_{p^\varepsilon}(v^{-1}(X_{\mathrm{in}}), v^{-1}(X_{\mathrm{out}})) \cap E_s(p^\varepsilon)\right| \ge c \log K$$

due to Assumption 3, and similarly for $v'$. Since $M \ge cK$, we may choose $C$ a large enough constant so that

$$C > \frac{n_{\mathrm{out}} d_{\mathrm{out}}}{c} \cdot \frac{K \log q}{q \log K}.$$

Then since

$$\left|\mathcal{M}_{v_\sharp p^\varepsilon}(X_{\mathrm{in}}, X_{\mathrm{out}}) \cap E_s(v_\sharp p^\varepsilon)\right|, \left|\mathcal{M}_{v'_\sharp p^\varepsilon}(X_{\mathrm{in}}, X_{\mathrm{out}}) \cap E_s(v'_\sharp p^\varepsilon)\right| > \left|E_s(v_\sharp p^\varepsilon) \cap E_s(v'_\sharp p^\varepsilon)\right|$$

it must hold that

$$\mathcal{M}_{v_\sharp p^\varepsilon}(X_{\mathrm{in}}, X_{\mathrm{out}}) \cap E_s(v_\sharp p^\varepsilon) \neq \mathcal{M}_{v'_\sharp p^\varepsilon}(X_{\mathrm{in}}, X_{\mathrm{out}}) \cap E_s(v'_\sharp p^\varepsilon).$$

Without loss of generality, we may suppose there exists a sparse edge $\xi = (X_{t-1}, X_t)$ of $v_\sharp p^\varepsilon$ that is included in the left-hand side of the above but not the right. Since $\mathcal{A}(\xi)$ is included in the computation of $r_{\mathcal{A}, v_\sharp p^\varepsilon}$ along $\mathcal{M}_{v_\sharp p^\varepsilon}(X_{\mathrm{in}}, X_{\mathrm{out}})$ but not of $r_{\mathcal{A}, v'_\sharp p^\varepsilon}$ along $\mathcal{M}_{v'_\sharp p^\varepsilon}(X_{\mathrm{in}}, X_{\mathrm{out}})$, denoting $\mathcal{A}^- := \mathcal{A}|_{S \times S \setminus \{\xi\}}$, it follows that

$$
\begin{aligned}
&\langle h_{v_\sharp p^\varepsilon}, h_{v'_\sharp p^\varepsilon} \rangle_{\mathcal{H}} \\
&= \mathbb{E}_{X_{\mathrm{in}}, X_{\mathrm{out}}, \mathcal{A}} \left[h_{v_\sharp p^\varepsilon}(X_{\mathrm{in}}, X_{\mathrm{out}}, \mathcal{A}) h_{v'_\sharp p^\varepsilon}(X_{\mathrm{in}}, X_{\mathrm{out}}, \mathcal{A})\right]
\end{aligned}
$$

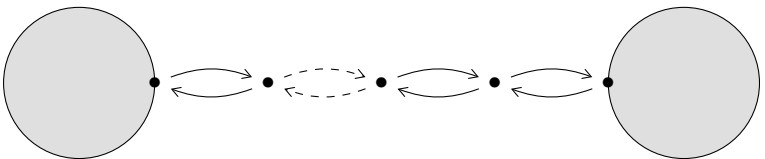

*Figure 2.* Sparse edge construction for the no-search scenario. The two circles represent the original dense clusters. Dashed edges have probability $\varepsilon$.

$$= \mathbb{E}_{X_{\text{in}}, X_{\text{out}}, \mathcal{A}} \left[ \phi \circ r_{\mathcal{A}, v_\sharp p^\varepsilon} (\mathcal{M}_{v_\sharp p^\varepsilon}(X_{\text{in}}, X_{\text{out}})) \phi \circ r_{\mathcal{A}, v'_\sharp p^\varepsilon} (\mathcal{M}_{v'_\sharp p^\varepsilon}(X_{\text{in}}, X_{\text{out}})) \right]$$

$$= \mathbb{E}_{X_{\text{in}}, X_{\text{out}}, \mathcal{A}^-} \left[ \mathbb{E}_{\mathcal{A}(\xi)|\mathcal{A}^-} \left[ \phi \circ r_{\mathcal{A}, v_\sharp p^\varepsilon} (\mathcal{M}_{v_\sharp p^\varepsilon}(X_{\text{in}}, X_{\text{out}})) \right] \phi \circ r_{\mathcal{A}, v'_\sharp p^\varepsilon} (\mathcal{M}_{v'_\sharp p^\varepsilon}(X_{\text{in}}, X_{\text{out}})) \right].$$

Adopting the shorthand $g1_A = g$ if $A$ holds and $g1_A = e_G$ if not, we may write

$$r_{\mathcal{A}, v_\sharp p^\varepsilon} (\mathcal{M}_{v_\sharp p^\varepsilon}(X_{\text{in}}, X_{\text{out}})) = (\mathcal{A}_L \circ \mathcal{A}(X_{t-1}, X_t) \circ \mathcal{A}_R) \cdot \psi(X_{\text{in}})$$

where

$$\mathcal{A}_L = \mathcal{A}(X_{T-1}, X_T) 1_{\{(X_{T-1}, X_T) \in E_s(v_\sharp p^\varepsilon)\}} \circ \cdots \mathcal{A}(X_t, X_{t+1}) 1_{\{(X_t, X_{t+1}) \in E_s(v_\sharp p^\varepsilon)\}},$$

$$\mathcal{A}_R = \mathcal{A}(X_{t-2}, X_{t-1}) 1_{\{(X_{t-2}, X_{t-1}) \in E_s(v_\sharp p^\varepsilon)\}} \circ \cdots \mathcal{A}(X_0, X_1) 1_{\{(X_0, X_1) \in E_s(v_\sharp p^\varepsilon)\}}.$$

Since $\mathcal{A}(X_{t-1}, X_t)$ is uniformly distributed on $G$, the composition $\mathcal{A}_L \circ \mathcal{A}(X_{t-1}, X_t) \circ \mathcal{A}_R$ is also uniformly distributed on $G$ conditioned on $\mathcal{A}^-$; note that this step relies crucially on the existence of left and right inverses in $G$. Hence

$$\mathbb{E}_{\mathcal{A}(\xi)|\mathcal{A}^-} \left[ \phi \circ r_{\mathcal{A}, v_\sharp p^\varepsilon} (\mathcal{M}_{v_\sharp p^\varepsilon}(X_{\text{in}}, X_{\text{out}})) \right] = 0$$

for all $X_{\text{in}}, X_{\text{out}}, \mathcal{A}^-$, so that $\langle h_{v_\sharp p^\varepsilon}, h_{v'_\sharp p^\varepsilon} \rangle_{\mathcal{H}} = 0$. Thus the subset $\{h_{v_\sharp p^\varepsilon} : v \in \mathcal{V}\}$ of $\mathcal{H}$ is orthogonal with size at least

$$A_q(K, \tau K) \geq q^{\frac{\log \log q}{2Cq} K} \geq K^{\Omega(\log \log K)}$$

if $q = \Theta(K)$, which grows faster than any polynomial in $K$. In the case that $q/K \to \infty$, we may choose a smaller $q' = cK$ at the beginning to obtain the same lower bound. Therefore we have shown that $\text{SDA}(\mathcal{P}; \varnothing) \geq K^{\omega(1)}$.

Finally, if the minimum length in Assumption 3 is instead set to scale linearly as $cK$ for some $0 < c < 1$, it suffices to set $C = 1$ and choose $q$ (equivalently $M$) a large enough constant satisfying

$$n_{\text{out}} d_{\text{out}} \frac{\log q}{q} < c$$

to apply the same argument. Then the lower bound (23) becomes exponential in $K$.

**Lower bounding** $\text{SDA}(\mathcal{P}; \mathcal{M})$. We take an arbitrary kernel $p \in \mathbb{P}$ and make the following modification, pictured in Figure 2. Each sparse edge $(x, y)$ is replaced by a set of states $z_0 = x, z_1, \cdots, z_{q-1}, z_q = y$ such that each neighboring pair $z_t, z_{t+1}$ for $t \in \mathbb{Z}_q$ are connected to each other via bidirectional edges of probability $O(1)$. However, one specific pair $z_t, z_{t+1}$ is to be connected to each other with probability $\varepsilon$. Denote this index by $v_j$ where $j \in [J]$ numbers the set of directly connected clusters; it holds that $J = |E_s| = \Theta(K)$ by Assumption 2. The points $z_t$ for $t \leq v_j$ and for $t > v_j$ are appended to the clusters containing $x$ and $y$, respectively. Since this is a bounded number of points all connected by constant probability edges, the extended clusters are still rapidly mixing. Then each vector $v = (v_j) \in \mathbb{Z}_q^J$ determines a kernel $p_v \in \mathcal{P}$, and

$$|E_s(p_v) \cap E_s(p_{v'})| = 2(J - d_H(v, v'))$$

holds for all $v, v' \in \mathbb{Z}_q^J$. We now repeat the argument from before to show orthogonality: by Lemma E.3 there exists a $\tau J$-separated subset $\mathcal{V} \subset \mathbb{Z}_q^J$ of size $A_q(J, \tau J)$ for $\tau = 1 - \log q/q$, and by choosing $q$ large enough we can ensure

$$|\mathcal{M}_{p_v}(X_{\text{in}}, X_{\text{out}}) \cap E_s(p_v)| \geq cK > \left( \frac{2 \log q}{q} \right) J \geq |E_s(p_v) \cap E_s(p_{v'})|.$$

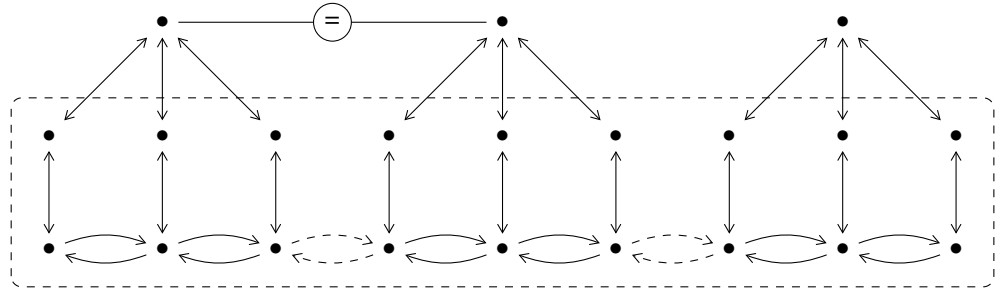

*Figure 3.* Graph construction for the local search scenario ($r = 2$). Dashed edges have probability $\varepsilon$. A local neighborhood of maximum distance one from the original graph is shown in the dashed box, which the learner is assumed to have full access to.

The key aspect of this construction is that the support of $p_v$ does not depend on $v$, as only the position of the sparse edge among $q$ candidates change with $v$. This implies that querying the path $\mathcal{M}_{p_v}(X_{\text{in}}, X_{\text{out}})$ (and indeed the entire set of edges) does not reveal any information about the ground truth $v$, and $\{p_v : v \in \mathcal{V}\}$ satisfies the conditions to compute the SQ dimension with access to $\mathcal{M}$. Hence we have shown that

$$\text{SDA}(\mathcal{P}; \mathcal{M}) \geq A_q(J, \tau J) \geq e^{\Omega(K)}.$$

**Lower bounding** $\text{SDA}(\mathcal{P}; \text{nbd}(\mathcal{M}))$**.** We construct the graph depicted in Figure 3 as follows. We start with $nK$ clusters of size $M$ laid out in side by side and connect neighboring clusters with bidirectional edges of probability $\varepsilon$. From all states extend a 'rod' of probability $O(1)$ bidirectional edges of bounded length $r$ (the vertically arranged states, here $r = 2$), similarly to Figure 2. Join the endpoints of all rods originating from each cluster into a single 'endpoint' state. For $\mathcal{D}$, we assume that $(X_{\text{in}}, X_{\text{out}})$ are sampled only from the original clusters (bottom horizontal line of states) and $\mathcal{M}_p$ only returns paths along this line.

Choose a size $K$ subset $B$ of the low probability edges to be sparse edges, viewed as a subset of $[nK]$. For each of the edges $(x, y)$ not in $B$, identify the endpoint states of the clusters containing $x, y$. The identified clusters will merge into a single larger rapidly mixing cluster, so that $(x, y)$ is indeed no longer a sparse edge. Denote the resulting kernel as $p_B$. We have the following intersection constraint bound:

**Lemma E.5.** *For any $n \geq 5$, there exists a size $e^{\Omega(K)}$ set $\mathcal{B}$ of size $K$ subsets of $[nK]$ such that $|B \cap B'| \leq cK$ for all $B, B' \in \mathcal{B}$.*

*Proof.* We construct $\mathcal{B}$ via a greedy algorithm similarly to the Gilbert-Varshamov bound. Start with $\mathcal{B} = \varnothing$ and add any $B \subset [nK]$ of size $K$ not already in $\mathcal{B}$ to $\mathcal{B}$. Each new element blocks at most

$$\binom{K}{cK}\binom{nK - cK}{K - cK}$$

elements from being added to $\mathcal{B}$. Hence the maximum size of $\mathcal{B}$ is at least

$$\binom{nK}{K}\binom{K}{cK}^{-1}\binom{nK - cK}{K - cK}^{-1} = \binom{nK}{K}\binom{K}{cK}^{-2} \geq O(n^K K^{-1/2}) \cdot 2^{-2K} \geq e^{\Omega(K)}$$

for $n \geq 5$. $\qquad\square$

Then $|E_s(p_B) \cap E_s(p_{B'})| = |B \cap B'| \leq cK$ and we can repeat the same argument to show orthogonality. Furthermore by taking $r$ sufficiently large, the local neighborhood of any path contained in the bottom horizontal line (which must be contained in the dashed area in Figure 3) is isomorphic for all $B \in \mathcal{B}$, since it cannot query the endpoint states to identify which clusters are actually connected. Hence $\{p_B : B \in \mathcal{B}\}$ satisfies the conditions to compute the SQ dimension with access to $\text{nbd}(\mathcal{M})$, and thus

$$\text{SDA}(\mathcal{P}; \text{nbd}(\mathcal{M})) \geq |\mathcal{B}| \geq e^{\Omega(K)}.$$

$\qquad\square$

