# OpenReview forum: "Metastable Dynamics of Chain-of-Thought Reasoning: Provable Benefits of Search, RL and Distillation"
_ICML.cc/2025/Conference — ICML 2025 poster_

### Official Review · Reviewer_R4eB · 2025-03-13

**Overall Recommendation:** 3

**Summary:**

This paper models Chain-of-Thought (CoT) reasoning as a metastable Markov process, where dense clusters correspond to easy reasoning steps and sparse edges represent difficult transitions. It provides a theoretical analysis of how search, reinforcement learning (RL), and distillation improve hitting times by identifying and enhancing sparse transitions. The study also employs statistical query (SQ) complexity arguments to establish the necessity of global search.

**Claims And Evidence:**

My only concern about the claim is that the use of "dynamics" to describe the transition process may lead to confusion with training dynamics of COT.

**Essential References Not Discussed:**

No

**Experimental Designs Or Analyses:**

There is no experiment.

**Methods And Evaluation Criteria:**

Yes

**Other Comments Or Suggestions:**

Please see above.

**Other Strengths And Weaknesses:**

### **Strengths**
1. The metastable Markov chain abstraction offers a principled framework for formalizing CoT inference dynamics, linking metastability theory (e.g., hitting times) to large language model (LLM) reasoning.
2. The analysis elucidates why inference-time compute (via search) and fine-tuning (via RL) enhance CoT by prioritizing sparse transitions, while distillation compresses cluster-level dynamics. The SQ lower bound provides a rigorous argument for the necessity of global search.

---

### **Weaknesses**
1. Modeling CoT as a Markov chain reduces reasoning to state transitions, potentially overlooking key aspects such as multi-step logical consistency and algorithmic operations.
2. While search, RL, and distillation improve hitting times in the model, the analysis does not establish a direct connection to CoT’s unique characteristics. The results could generalize to any sparse transition system and efficient edge-search algorithm, raising questions about the theory’s specificity to CoT reasoning.

**Questions For Authors:**

Please see above.

**Relation To Broader Scientific Literature:**

The metastable Markov chain abstraction offers a principled framework for formalizing CoT inference dynamics, linking metastability theory (e.g., hitting times) to large language model (LLM) reasoning.

**Theoretical Claims:**

The proof seems correct.

---

> ### Author Rebuttal · Authors · 2025-03-26
>
> Thank you for your detailed review and suggestions! Our responses are as follows.
>
> **Claims And Evidence** - Thank you for pointing out the possible confusion. We have made sure that the training dynamics is only referred to as "optimization guarantee", "training dynamics", "gradient descent iterates", etc.
>
> **Response to Weakness 1** - While this is true for the basic Markov model in Section 2, the logical reasoning task we introduce in Section 5 is an example of how this model can be easily enriched to include notions of correctness or algorithmic computation. The sequential nature of the logical computation task introduced in line 362 makes it so that each step of the chain must be correct in order to obtain the final answer, which is an intuitive property of CoT and is key to the proof of the SQ lower bound - if any single step is wrong, the final output will be uniformly distributed and contain no useful information.
>
> For a simple application, suppose the model must execute the logical computation at each step, with a small probability of failure $p$. An immediate corollary of decreased hitting time (Theorem 3.2) is that the overall probability of failure is improved from $1-(1-p)^{|S|/\varepsilon}$ to $1-(1-p)^{|S|/\varepsilon_{\text{max}}}$. Hence while a priori longer CoT can help in reaching target states which are farther away, RL can produce shorter, more efficient CoT for each target, likely resulting in less error accumulation or hallucinations.
>
> Nonetheless, we have elected to focus on the Markov model for the majority of the paper and only introduce the logic task towards the end to prove the lower bounds, for simplicity of presentation.
>
> Another natural extension to incorporate the multi-step nature of CoT is to model reasoning as a Markov decision process rather than a Markov chain, as in [SRLK25]. That is, the full state space $S^*=\cup_{t\ge 1} S^t$ is the set of trajectories $(s_1,\cdots,s_t)\in S^t$, and each new action (transition) $s_t\to s_{t+1}$ is concatenated with the previous trajectory to give the next state $(s_1,\cdots,s_{t+1})$. This subsumes our model by relaxing the memorylessness assumption, which is more realistic since CoT should take into account all previous reasoning steps. Then we can again study clustering behavior and metastable dynamics of the final token $s_t$ in the original state space $S$ under a more sophisticated base model/policy.
>
> **Response to Weakness 2** - In this paper, we devoted our attention to the idea of modeling easy versus hard steps, which led other aspects of the setup being quite abstract/general. However, we agree that more could be derived by studying a concrete class of reasoners. A natural extension to incorporate more CoT-specific behavior is to model the process as an MDP, as described above. This can then be used to study CoT dynamics for specific tasks, for example, the parity task which has been used to show the computational benefits of CoT optimization [KS24] can be understood by taking the state space to be the set of binary strings, and reasoners as executing 2-parities at each step.
>
> [SRLK25] Scaling test-time compute without verification or RL is suboptimal. Amrith Setlur, Nived Rajaraman, Sergey Levine, Aviral Kumar. arXiv preprint arXiv:2502.12118.
>
> [KS24] Transformers provably solve parity efficiently with chain of thought. Juno Kim, Taiji Suzuki. ICLR 2025.
>
> [WZLZ24] From sparse dependence to sparse attention: unveiling how chain-of-thought enhances transformer sample efficiency. Kaiyue Wen, Huaqing Zhang, Hongzhou Lin, Jingzhao Zhang. ICLR 2025.

---

### Official Review · Reviewer_XmAg · 2025-03-14

**Overall Recommendation:** 4

**Summary:**

The authors propose a theoretical analysis of Chain-of-Thought (CoT) reasoning. Using a graph theoretical approach, they distinguish “easy” reasoning steps as intra-dense cluster connections from harder reasoning steps, which connect different clusters but have low probability of being picked at inference. They analyse a search methodology aiming to increase the probability of transitioning between different clusters (i.e. of CoT exploring harder reasoning steps) in order to improve the efficiency of reasoning. This framing also helps understand how reasoning can be distilled into a simpler model. Finally, this work shows the limitations of reasoning in some settings without global search.

## update after rebuttal
I maintain my recommendation to accept the paper.

**Claims And Evidence:**

Claims are proven in the appendix based on the hypotheses. I think the article does a good job of targeting timely concepts in the field that may seem intuitive but that lacked theoretical grounding.

**Essential References Not Discussed:**

I think the paper discussed existing literature decently well considering the page constraints.

**Experimental Designs Or Analyses:**

This is a theory paper.

**Methods And Evaluation Criteria:**

This is a theory paper.

**Other Comments Or Suggestions:**

Typos:
1. “We use sign gradient descent for simplicity of analysis (ordinary gradient descent” should be “ascent” (since you are solving a maximisation problem).

**Other Strengths And Weaknesses:**

Generally, I would have liked to see a practical application / outcome of this theoretical analysis. Still, I do understand that there is value in having a theoretical framing/results of intuitive notions, which may help the community build practical methods inspired from this theory.

**Questions For Authors:**

1. Prop 2.1: what is meant by “M->\infty”? I assume it’s the definition given in 2.1 of M that should be used here, but if that’s the case I suggest switching “M” for something else in the paragraph preceding prop 2.1 as there, M is a set (more specifically, a subset of a finite state space).
2. P7: what would be the inverse element in the group G with the example that you gave?

**Relation To Broader Scientific Literature:**

This paper proposes a theoretical framework for recent methodologies to improve reasoning performance of foundation models, generally based on increasing compute at inference.

**Theoretical Claims:**

Proofs are provided for the theoretical claims, but **see important caveat about supplemental material**.

---

> ### Author Rebuttal · Authors · 2025-03-26
>
> Thank you for your positive assessment of our contributions and helpful suggestions! Our responses are as follows; typos have been fixed.
>
> **Response to Weakness** - While we did not conduct any experiments due to time and compute constraints, the formulation and theory do give quantitative predictions that could be empirically studied in a future work. For example, we can measure the length of CoT to solve difficult math problems as a function of the amount of finetuning to validate Theorem 3.2. Another approach is to train a “difficulty” model to score how easy or hard each reasoning step is, and probe the observed sparse-dense dynamics of CoT (e.g. by looking at the density and distribution of hard steps before/after finetuning).
>
> We also acknowledge that the practical value of our model is limited insofar as it is quite abstract and simplified; please see our rebuttal to Reviewer R4eB on directions to improve this.
>
> **Questions**
> 1. In Prop 2.1, $M$ refers to the cluster size $M:=\max_k |C_k|$ of our setup, not the set $M$. We agree the notation is confusing and will replace the set $M$ with $\mathcal{M}$.
> 2. When $\mathcal{R}=G=\mathbb{Z}_2$, the identity element is 0. When $G$ is a space of functions, the identity element is the identity function. In the latter example, functions in $G$ must be necessarily invertible (bijective) to form a group rather than a monoid.

---

> > ### Comment · Reviewer_XmAg · 2025-04-06
> >
> > Thank you for your response and answering my questions, and believe updating the paper with those answers will be appreciated by your readers.
> >
> > As mentioned, I fully understand that theoretical work is inherently useful to the community in that it may precisely lead to future work deriving empirical results from it, so I do not think it's a "weakness" when it comes to accepting the paper.

---

### Official Review · Reviewer_QuhA · 2025-03-14

**Overall Recommendation:** 4

**Summary:**

This paper develops a theoretical framework for understanding chain-of-thought (CoT) reasoning in large language models by modeling it as a metastable Markov process. They prove that implementing search protocols that reward sparse edges improves reasoning by decreasing the expected steps to reach different clusters. The authors show that a compressed metastable representation of the reasoning dynamics can be distilled into a smaller, more efficient model that preserves essential properties of the original chain.

**Claims And Evidence:**

The claims in this paper are generally well-supported by mathematical proofs and theoretical analysis.

**Essential References Not Discussed:**

-

**Experimental Designs Or Analyses:**

The paper is entirely theoretical with no experimental validation.

**Methods And Evaluation Criteria:**

The paper is entirely theoretical with no experimental validation.

**Other Comments Or Suggestions:**

See weaknesses.

**Other Strengths And Weaknesses:**

Strengths:
1. Proposes a theoretical framework that combines Markov process theory with language model reasoning, providing new insights into reasoning challenges.
2. Provides principled explanations for empirically observed phenomena in LLM reasoning capabilities.

Weaknesses:
1. Some assumptions about metastable dynamics may be idealized compared to actual reasoning patterns.

**Questions For Authors:**

See weaknesses.

**Relation To Broader Scientific Literature:**

-

**Theoretical Claims:**

I check several key proofs including Theorem 3.2, Propositions 3.3-3.4, and Theorem 4.3. The proofs appear mathematically sound with no substantial errors.

---

> ### Author Rebuttal · Authors · 2025-03-26
>
> Thank you for your positive assessment of our contributions! Regarding the simplicity of the Markov model compared to actual reasoning, please see our rebuttal to similar weaknesses pointed out by Reviewer R4eB.

---

### Decision · Program_Chairs · 2025-05-01

**Decision:**

Accept (poster)

**Comment:**

This theoretical work addresses the problem of finding high-reward trajectories from an LLM models. Posing the "steps" (sentences or expressions) generated by an LLM as nodes in a graph, it proves that biasing the search towards "sparse" edges that connect different clusters shortens the search time.

All reviewers favor the acceptance of this work, which provides direct guidance for a more efficient use of test-time compute. The provided results are novel and seem correct, and provide a principled explanation of phenomena observed in practice in LLM. This work would however be strengthened by the inclusion of experimental results beyond the provided proofs.